# Low confidence in multi-decadal trends of wind-driven upwelling across the Benguela Upwelling System

Mohammad Hadi Bordbar, Volker Mohrholz and Martin Schmidt

Leibniz Institute for Baltic Sea Research Warnemünde (IOW), Rostock, Germany

*Correspondence to*: Mohammad Hadi Bordbar (hadi.bordbar@io-warnemuende.de; Ocean.Circulation@gmail.com)

**Abstract.** Like other Eastern Boundary Upwelling Systems, in the Benguela Upwelling System, the upwelling along the coastline is primarily alongshore-wind-driven. In contrast, it is mainly driven by the wind stress curl farther offshore. The surface wind regime across the Benguela Upwelling System is strongly related to the South Atlantic Anticyclone that is believed to migrate poleward in response to anthropogenic global warming. Using the European

Centre for Medium-Range Weather Forecasts ERA5 reanalysis for 1979-2021, we investigate multi-decadal changes of the South Atlantic Anticyclone and their impacts on coastal upwelling driven by alongshore winds, wind-stress-curl-driven upwelling within the coastal zone and total upwelling as the sum of both across the Benguela Upwelling System. Even though the detailed structure of surface wind over the coastal zone matters for both alongshore-wind-driven coastal upwelling and wind-stress-curl-driven upwelling, we show that it is not of major importance for the total

amount of upwelled water. We found a robust connection between the anticyclone intensity and the zonally integrated wind-stress-curl-driven and total upwelling. However, such connection for the alongshore-wind-driven coastal upwelling is weak. The upwelling in the equatorward portion of the Benguela Upwelling System is significantly affected by the anticyclone intensity. In contrast, the poleward portion is also influenced by the meridional position of the anticyclone. In general, the impacts of the anticyclone on the local upwelling are more robust during the austral

winter. The multi-decadal trend in the sea level pressure across the South Atlantic renders a considerable heterogeneity in space. However, this trend features a small signal-to-noise ratio and can be obscured by interannual to decadal climate variability. This view is further supported by the coastal and wind-stress-curl-driven upwelling in several upwelling cells showing hardly any significant multi-decadal trends.

## 1 Introduction

Human-induced changes in the Earth's climate system have raised enormous concern for the future of marine ecosystems across the major Eastern Boundary Upwelling Systems (EBUS) with considerable resources of the world's pelagic fish (Pauly and Christensen, 1995; Rykaczewski et al., 2015; Abraham et al. 2021). At the same time, modes of natural climate variability, which span a broad range of timescale exert profound impacts on the functioning of the EBUS marine ecosystems (Jarre et al., 2015; Rykaczewski et al., 2015; Yari et al., 2023). These modes result from the

interactions between different components of the Earth's climate system (i.e., atmosphere, ocean, cryosphere, etc.) and modulate the ocean-atmosphere heat, mass, and momentum exchanges on the regional or even global scale.

The Benguela Upwelling System (BUS) is located at the eastern margin of the subtropical South Atlantic and extends from the Cape of Good Hope in South Africa to southern Angola (Shannon 1985). The BUS is one of the world's most productive marine ecosystems with several distinct upwelling cells (Johnson, 1976; Kainge et al., 2020;

Jarre et al., 2015). In each cell, the wind stress and its spatial heterogeneity are major upwelling drivers (Fennel, 1999).

Near the coast, upwelling is proportional to the alongshore wind stress and associated with offshore-directed transport. This type of upwelling is referred to as coastal upwelling. It is vigorous but confined to narrow coastal zones (~30 km wide). Farther offshore, the Ekman transport shows a relatively weak divergence resulting in an upward velocity that is proportional to the local negative wind stress curl. This source of upwelling is termed the wind-stress-curl-driven

(WSCD) upwelling. Typically, the WSCD upwelling is much weaker than the coastal upwelling, but extends over a broader area (Fennel, 1999; Bordbar et al., 2021). Given the different characteristics of these two sources of upwelling, they favor different pelagic food webs (Rykaczewski et al., 2008; Lamont et al., 2019).

A steep drop-off in the alongshore wind towards the coast often occurs near the coastal zones of the BUS, driving an intensified nearshore WSCD upwelling. From an analytical theory of upwelling and results of a state-of-the-art

ocean circulation model, Bordbar et al. (2021) showed that the volume of upwelled water due to the coastal offshore transport and the wind stress curl is almost in the same order of magnitude in the BUS.

Feistel et al. (2013) showed that Namibian shelf upwelling events are closely connected to the sea level pressure (SLP) changes in St. Helena Island (5.7°W, 15.95°S). Based on observed surface air temperature, SLP, and precipitation, they introduced the St. Helena Island Climate Index to describe interannual anomalous coastal warm and

cold events, known as Benguela Niños and Niñas. Warm events are associated with the southward intrusion of low-oxygenated and nutrient-enriched equatorial waters with severe consequences for marine ecosystems (Brandt et al., 2023). Anomalies in the local wind stress and associated coastal upwelling along the southwest African coast and the relaxation of trade winds over the western equatorial Atlantic, which excites downwelling Kelvin waves propagating eastward along the equator and thereafter southward along the west African coast, play central roles in the onset and

development of the Benguela Niños (Shanon et al., 1986; Richter et al., 2010; Lubbecke et al., 2010; Brandt et al., 2023). These wind anomalies are often associated with slowing down in the subtropical South Atlantic Anticyclone (SAA).

Surface wind across the Namibian shelf is affected by a northward atmospheric low-level jet, which is known as the Benguela low-level coastal jet. The signatures of this low-level jet are more prominent at about 17°S and 25°S-

30°S, which coincide with Kunene and Lüderitz upwelling cells, respectively (Patricola and Chang, 2017; Brandt et al., 2023). In general, the atmospheric flow across the subtropical South Atlantic is largely influenced by the SAA (Feistel et al., 2003; Richter et al., 2008; Lamont et al., 2018). From the northernmost part (~16°S) of the BUS to the north of Lüderitz (~27°S), termed the equatorward part of the BUS (nBUS), the surface wind persists year-round and is approximately proportional to the cross-shore SLP gradient between the SAA and the Angola-Kalahari low-pressure

system (Feistel et al., 2003). From the south of Lüderitz (~27°S) to the southernmost part of the BUS (~33°S), referred to as the poleward portion of the BUS (sBUS), the wind undergoes strong seasonal variations (Shannon 1985; Shannon and Nelson 1996). The total amount of upwelled water associated with coastal offshore transport in the nBUS is known to be affected by the strength of the SAA (Jarre et al., 2055; Lamont et al., 2018). Enhanced coastal offshore transport was observed in the sBUS when the SAA shifted to the south. However, it is still unclear whether the coastal upwelling

across the nBUS is significantly influenced by the meridional displacement of the SAA (Jarre et al., 2055; Lamont et al., 2018).

The SLP variability across the South Atlantic is affected by natural modes of climate variability. The Southern Annular Mode (SAM) is the dominant mode of climate variability in the extratropical southern hemisphere and is expressed as

a ring-shaped structure of the SLP anomalies around the polar latitudes with fluctuations ranging from synoptic to decadal timescales (Gilett et al. 2006; Wachter et al. 2020; Fogt and Marshall 2020). The positive phase of the SAM is defined as a positive anomaly in the meridional pressure gradient between relatively low pressure located over the southern hemisphere polar latitude and high pressure at the mid-latitudes. Over recent decades, the SAM has undergone a positive trend (Wachter et al. 2020). The El Niño Southern Oscillation, the Atlantic Niño, the interdecadal Pacific Oscillation, and the Atlantic Multidecadal variability are examples of internal modes of climate variability affecting the South Atlantic climate (Shannon et al., 1986; Kidson, 1988; Gillett et al., 2006; Sun et al. 2017; Rouault & Tomety, 2022). For example, using ERA-interim and JRA-55 reanalysis data sets, Sun et al. (2017) showed that La Niña events coincide with a poleward migration of the SAA, and almost opposite displacement is observed during El Niño events. The future of wind-driven upwelling across the subtropical eastern edge of major ocean basins inspired several studies (Bakun, 1990; Narayan et al., 2010; Sydeman et al., 2014; Rykaczewski et al., 2015). Using observational wind products over the last decades, Lamont et al. (2018) and Abrahams et al. (2021) found a significant downward trend in the number of offshore-directed coastal Ekman transport events across the nBUS, whereas it underwent an upward trend in the sBUS. The majority of climate models project an acceleration (slight deceleration) of upwelling favorable winds over the poleward (equatorward) margins of the EBUS (Rykaczewski et al., 2015; Bonino et al., 2019). However, these simulated trends were less prominent in the BUS (Rykaczewski et al., 2015). These studies were inspired by a conceptual hypothesis raised by Bakun (1990), suggesting coastal upwelling would strengthen in response to global warming. The basic premise of this hypothesis comes back to the intensified cross-shore SLP gradient associated with excess warming over the landmass relative to adjacent ocean waters. During summer, when solar radiation reaches its seasonal maximum, the cross-shore SLP contrast enhancement is expected to be more severe.

Several limiting factors hinder the assessment of Bakun's hypothesis in the BUS. First, observations over this part of the South Atlantic are sparse in time and space. Second, climate models, widely used for past and future climate changes, suffer from a long-standing sea surface temperature (SST) bias over the southeast Atlantic with considerable impacts on the regional atmospheric flow (Sun et al., 2017; Li et al., 2020). Third, changes associated with internal modes of climate variability can overshadow the signature of the externally-forced trends (Bordbar et al., 2015; Tim et al., 2015; Latif et al., 2016; Bonino, 2019; Polonsky and Serebrennikov, 2021; Yari et al., 2023). For example, it is unclear how long it would take for incremental changes in the coastal wind to emerge from the background fluctuations associated with internal climate variability. One should keep in mind that the characteristics of the internal climate variability (i.e., magnitude, frequency) might change in response to enhanced radiative forcing.

Hence, it remains controversial whether the mechanism suggested by Bakun is the dominant factor for upwelling changes across the BUS. The major concern is that the WSCD upwelling is of great importance for marine ecosystems across the BUS, but neither its mechanism nor its response to global warming is considered in Bakun's hypothesis. It is important to bear in mind that the coastal and the WSCD upwelling do not necessarily undergo identical fluctuations and are sometimes out of phase (Rykaczewski et al., 2008; Bordbar et al., 2021). To understand the relation of the SAA with the coastal and the WSCD upwelling in the BUS, we assess their linkage from the ERA5 products over 1979-2021. We mainly analyze the variability of atmospheric quantities and associated impacts on upwelling across the BUS. We introduce the concept of potential upwelling to distinguish the quantities used in our analysis from those describing realistic, highly complex vertical transport processes in the ocean. In this sense, potential curl-driven

upwelling is the upwelling that would take place within an unbounded ocean with only wind-driven surface currents under the absence of other drivers of upwelling, baroclinicity, bottom topography, coastlines, geostrophic flow, inertial or planetary waves. Potential coastal upwelling characterizes an upwelling process driven solely by the alongshore wind and is related only to the cross-shore divergence of the wind-driven cross-shore directed flow. We derive the potential upwelling quantities from analytical theories of ocean dynamics, the steady state Ekman theory, and a theory of coastal upwelling given by Fennel (1999). This keeps the focus on well-defined quantities, even if they do not reflect realistic upwelling that may be modified by other processes like alongshore wind variability, coastally trapped waves, frontal dynamics, etc. We discuss the robustness of the long-term trends in the SAA, and the SLP over the South Atlantic. Furthermore, we assess the long-term changes in the probability distribution of coastal and WSCD upwelling in several coastal upwelling cells across the BUS in 1979-2021.

## 2 Data and methods

The hourly SLP and surface wind vectors from the European Centre for Medium-Range Weather Forecasts (ECMWF) ERA5 reanalysis for 1979-2021 (43 years) are analyzed in this study (Hersbach et al., 2018). The spatial resolution of the datasets is $0.25° \times 0.25°$ regular grid. The daily and monthly averages are computed from the hourly values.

The outputs of the models used in the ERA5 reanalysis are assimilated with a large set of observational records to produce spatially and temporally consistent data evolving closely with the observation. HadISST2 data set, which was developed by the UK Met Office Hadley Centre, is widely implemented in ERA5 reanalysis (Hirahara et al., 2016). HadISST2 is on a $0.25° \times 0.25°$ regular grid and is derived from in-situ observations and two infrared radiometers, including the Along Track Scanning Radiometer (ATSR) and the Advanced Very High-Resolution Radiometer (AVHRR) (Hirahara et al., 2016). From mid-2008 onward, OSTIA SST from the UK Met Office with a resolution of $0.05° \times 0.05°$ was also used in ERA5 reanalysis (Hirahara et al., 2016). OSTIA is based on various types of observation, including in-situ observation, geostationary satellites and microwave imagers. In the Agulhas region located on the southern border of the BUS, both OSTIA and HadISST successfully represent the sub-mesoscale eddies. Furthermore, co-variability of SST and wind over this region is well represented in ERA5 data (Hirahara et al., 2016). The connection of the SAA variability with the onset and development of Atlantic Niño events and associated SST changes over the Angola Benguela front (15°S-17°S) is well represented in the ERA5 data (Prigent et al., 2020). The poleward (equatorward) displacement of the SAA during austral summer when the tropical Pacific features La Niña (El Niño) events is well represented in the ERA5 data sets (Rouault & Tomety, 2022). In addition, the spatial structure of the SAM and meridional wind anomalies over the southern hemisphere during different SAM phases are well reproduced in the ERA5 data sets (Marshall et al., 2022).

However, the ERA5 data have some data uncertainty due to the relative accuracy of the observation, sparsity of the observation in time and space, systematic model biases, and data assimilation (Hersbach et al., 2018). The uncertainty is not homogenous in time and space. It becomes larger backward over time, particularly in the pre-satellite era when the quality of the observation was poor compared with present-day observation. There is another source of data uncertainty over coastal areas with frequent upwelling events. The offshore transport of upwelled cold water often forms SST fronts near the coast (de Szoeke and Richman 1984), altering the local structure of the wind stress curl. On the cold side of the front, the near-surface air column is stabilized, decelerating the local wind speed. At the same time,

the air column is destabilized over the warm side, and the wind intensifies (Chelton et al., 2004). In this way, small-scale SST fronts drive local convergence and divergence of the surface wind, which is proportional to the size of the crosswind SST gradient. This aspect of small-scale (i.e., sub-mesoscale) ocean-atmosphere interaction is not adequately represented in ERA5 reanalysis because the model's resolution is too coarse. Further, orographic features near the coast (i.e., mountain passes, coastline geometry, and capes), which are not well resolved in the model used in ERA5 reanalysis, can diverge or converge the winds locally and alter the structure of wind stress curl (Chelton et al., 2004).

To examine the accuracy of ERA5 data over the BUS, we utilize several observations, including satellite-derived daily ASCAT surface winds covering 2007-2021 (Ricciardulli and Wentz, 2016), in situ SLP measurements in St. Helena Island (5.7°W-15.95°S) with a long-term record from 1893 to the present (Feistel et al., 2003). We compare the observed and simulated fluctuations on monthly to decadal time scales. We also compare the long-term trends in the observations and the ERA5 reanalysis. Further, we qualitatively examine the ERA5 SLP by using the time series of observed surface pressure in several climate stations near the southwest African coast, including Luanda (13.25°E, 8.85°S), Benguela (13.42°E,12.58°S), Port Nolloth (16.87°E,29.23°S), and Cape Town (18.60°E,33.96°S). For a given weather regime, the SLP in a station represents the surface air pressure if the station was located at the altitude of global mean sea level. The elevation of the selected climate stations from the global mean sea level is typically less than 80 meters; therefore, it is reasonable to assume that the observed surface pressure closely follows the SLP time series.

Consistent with a previous study (Belmonte Rivas and Stoffelen 2019), ERA5 for the southwest African coast agrees well with available observation-based data sets (Supplementary Info; Fig. S1-6). For St. Helena Island, the time series of the monthly, yearly, and 11-year running SLP means obtained from ERA5 evolved very closely with those derived from the observation (Fig. S1). The time series of simulated and observed yearly mean SLPs display only marginal trends, which are about 0.0025 and -0.0056 hPa/year, respectively. The available time series of surface pressure observed at Luanda, Benguela, Port Nolloth, and Cape Town climate stations evolve closely with the EAR5 SLP (Fig. S2). However, the surface pressure time series for all stations is not continuous or sometimes appear to have offset accuracy issues (e.g., Benguela; see Fig. S2), which do not allow the evaluation of long-term trends. The spatial pattern of the ERA5 and ASCAT meridional wind trends for 2008-2021 agree reasonably well (Fig. S6). Overall, meridional wind near the coastal area of the BUS underwent an upward trend over the last decade. The spatial structure of the trend in the model appears to be smooth relative to that derived from ASCAT wind. In addition, the wind intensification south of 30°S is pronounced in the ASCAT winds, whereas a relatively small trend is derived from ERA5 data.

To compute the wind stress, $\tau$, we use a bulk formula as

$$\vec{\tau} = C_d \rho_a U_{10} \vec{U}_{10}, \tag{1}$$

where $U_{10}$, $\vec{U}_{10}$, $\rho_a$ and $C_d$ represent the wind velocity magnitude (m/s) at 10-meter height above the sea surface, the surface wind velocity, surface air density (kg/m$^3$), and the dimensionless neutral drag coefficient, respectively. $\rho_a$ and $C_d$ are taken as constant values at 1.23 kg/m$^3$ and 0.0013 assuming neutral stability in the atmospheric boundary layer (Gill 1982).

The Ekman wind-driven ocean current theory is broadly used to describe the flow at the ocean surface. It is based on the balance between the vertical flux of horizontal momentum associated with wind stress, $\tau$, and the Coriolis force (Ekman 1905). In this theory, Ekman zonal ($U_E$) and meridional ($V_E$) volume transport per unit length (m²/s) are expressed as:

$$U_E = \frac{\tau_y}{\rho_w f} \qquad and \qquad V_E = \frac{-\tau_x}{\rho_w f} \ , \tag{2}$$

where $\rho_w$ and $f$ are the density of seawater and Coriolis parameter, respectively. $f$ is negative in the Southern Hemisphere. The divergence of the Ekman transport in the open ocean, which is proportional to the wind stress curl, is related to a vertical velocity in the water column. The WSCD upwelling in the f-plane approximation (i.e., invariant $f$) (Johnson 1976; Fennel and Lass 2007) is:

$$w_{curl} = \frac{1}{\rho_w f}\left(\frac{\partial \tau_y}{\partial x} - \frac{\partial \tau_x}{\partial y}\right). \tag{3}$$

Since the orientation of southwest African coastlines is almost in the meridional direction and the major component of the wind stress orients meridionally, we take the meridional component as the alongshore wind stress (Fennel 1999; Bordbar et al., 2021). In this way, the major element of the Ekman transport is the zonal component, taken as a good approximation for the cross-shore component. The major contributor to the wind stress curl is the zonal variation of the meridional wind stress. Hence, the WSCD (i.e., $w_{curl}$) upwelling can be approximated as:

$$w_{curl} \approx \frac{1}{\rho_w f}\left(\frac{\partial \tau_y}{\partial x}\right). \tag{4}$$

The balance between wind stress and Coriolis force from the Ekman transport, as the primary assumption of the Ekman dynamics, is disturbed near the coast. A downwind swift ocean current, known as coastal jet, forms near the coast (Yoshida 1959, Fennel 1999). In addition to the cross-shore directed Ekman transport another cross-shore directed transport component emerges, referred to as $U_c$. This way, the boundary condition of no flow through the coast is satisfied by the total cross-shore transport, $U_E + U_c$. The coastal upwelling associated with the divergence of the total cross-shore directed transport is confined to a coastal stripe with a width of about the first baroclinic Rossby radius of deformation ($R_1$) (Yoshida 1959, Fennel 1999). We approximate the coastal upwelling velocity ($w_{coast}$) as suggested in Bordbar et al. (2021) as:

$$w_{coast} = \frac{-2\tau^y(x=0)\, e^{\frac{2x}{R_1}}}{f \rho_w}\frac{1}{R_1}. \tag{5}$$

Here, $x$ is the distance to the coast. Note that x is negative in the westward offshore direction. $w_{coast}$ is reduced sharply with coastal distance and is negligible beyond the coastal distance of about $R_1$. The WSCD upwelling velocity, $w_{curl}$, is typically one order of magnitude smaller than the coastal upwelling velocity, $w_{coast}$. In turn, the coastal upwelling is localized within a narrow coastal stripe of a few 10km width. In contrast, $w_{curl}$ extends from the coast to a few 100km offshore.

Ocean dynamics is associated with many other flow elements, such as the formation of horizontal pressure gradients from upwelling, coastal jets, thermal fronts, sub-mesoscale instabilities, etc. (Fennel 1999; de Szoeke and Richman 1984; Abrahams et al., 2021). For example, the presence of geostrophic onshore directed current can remarkably alter the structure of coastal upwelling (Marchesiello and Estrade, 2010). When the surface and deep Ekman layers overlap

over shallow continental shelves, the cross-shore width of coastal upwelling is proportional to the inverse of the bottom slope (Marchesiello and Estrade, 2010). The occurrence of wind drop-off over this zone can cause a large divergence in offshore transport which can significantly alter the structure of coastal upwelling. Despite the importance of the processes above on the local scale, using a high-resolution ocean model, Bordbar et al. (2021) showed that the dominant driver of upwelling near the coast is the divergence of offshore transport. In addition, they showed that the WSCD

upwelling could be a reliable estimation of the total amount of upwelled water offshore. This motivates the choice of the atmosphere variables.

In this study, we address three potential upwelling quantities. First, we consider the cross-shore directed (i.e., zonal) integral of both upwelling contributions, which may be both of similar magnitude. For a location, x, in a distance from the coast much larger than $R_1$, (i.e., $|x| \gg R_1$), this integral reads:

$$W_{total}(x) = \int_x^0 dx'(w_{curl} + w_{coast}) \approx -\left(\frac{\tau^y(x)}{f\rho_w} - \frac{\tau^y(x=0)}{f\rho_w}\right) - \frac{\tau^y(x=0)}{f\rho_w} = -\frac{\tau^y(x)}{f\rho_w} = -U_E(x). \quad (6)$$

The accumulated amount of upwelled water, $W_{total}$, driven by the meridional wind between the coast and the position $x$ is finally transported offshore with the zonal Ekman transport. In this study, the integrals were carried out from the coast up to a point far offshore where the long-term average of wind stress curl equals zero, i.e., where the meridional wind stress is maximum (Fig. 1b). The second and the third potential upwelling quantities are cross-shore integrated

WSCD upwelling and coastal upwelling velocities, here referred to as $W_{curl}$ and $W_{coast}$, respectively. These quantities are estimated as follows:

$$W_{curl} = -\frac{\tau^y(x)}{f\rho_w} + \frac{\tau^y(x=0)}{f\rho_w} \quad (7)$$

$$W_{coast} = -\frac{\tau^y(x=0)}{f\rho_w}.$$

For the total (integrated) amount of upwelled water, i.e., $W_{total}$, details of the spatial pattern of the wind over the coastal

zones do not play a significant role. This is important for the coastal wind drop-off known to occur within a few 10-km coastal bands. It cannot be adequately resolved in the available data ERA5 and ASCAT data defined on a coarse (i.e., 0.25°×0.25°) grid. However, for both $w_{curl}$ and $w_{coast}$, those details matter. Underestimation of the coastal wind results in underestimated $w_{coast}$ and overestimated $w_{curl}$ and vice versa. However, this issue does not play a significant role in $W_{total}$. A summary of the quantities used to estimate the variation of wind-driven upwelling is given in table 1.

**Table 1: A summary of potential upwelling quantities (i.e., $W_{total}$, $W_{coast}$, $W_{curl}$), alongshore wind-driven upwelling, and wind-stress-curl-driven upwelling, which are used in this study. Positive values indicate upward velocity (i.e., upwelling) for all quantities.**

| Acronyms | Definition | Formula | Acronyms | Definition | Formula |
|---|---|---|---|---|---|
| $W_{curl}$ | Cross-shore integral of wind-stress-curl-driven upwelling velocity | $-\frac{\tau^y(x)}{f\rho_w}$ $+\frac{\tau^y(x=0)}{f\rho_w}$ | $w_{curl}$ | Wind-stress-curl-driven upwelling velocity | $\frac{1}{\rho_w f}\left(\frac{\partial\tau_y}{\partial x} - \frac{\partial\tau_x}{\partial y}\right)$ |
| $W_{coast}$ | Cross-shore integral of coastal upwelling velocity | $-\frac{\tau^y(x=0)}{f\rho_w}$ | $w_{coast}$ | Alongshore-wind-driven coastal upwelling velocity | $\frac{-2\tau^y(x=0)\,e^{\frac{2x}{R_1}}}{f\rho_w \quad R_1}$ |
| $W_{total}$ | Cross-shore integral of total upwelling velocity | $-\frac{\tau^y(x)}{f\rho_w}$ | | | |

In this study, we use the nearest grid point to the coast for computing the $W_{coast}$ and $w_{coast}$. We will discuss long time series of the potential upwelling quantities computed for the Kunene, Walvis Bay, Lüderitz, and Cape Columbine upwelling cells. To have a rough estimate of $w_{coast}$ in the upwelling cells, we compute the maximum upwelling velocity (see equation 5). We used $R_1$ from Chelton et al. (1998) in the nearest grid point to the upwelling cells (Fig. S7). We also compute the WSCD upwelling, i.e. $w_{curl}$, in each grid point (Fig. 1; see equation 3).

Mid-latitude atmospheric dynamics are characterized by frontal passages and passing cyclones and anticyclones shifting the position of semi-permanent SAA every several days (Richter et al., 2008; Gilliland and Keim, 2017; Sun et al., 2017). To identify the SAA center, we use monthly mean SLP values to filter out rapidly-varying migrating anticyclones, cyclones, and fragmented pressure systems. We employ a straightforward approach in the previous study to estimate the mean position and intensity of SAA (Gilliland and Keim 2017). First, we calculated the spatial average of the monthly-mean SLP in the region between 40°W-20°E and 45°-10°S. In the next step, the grid points with SLPs smaller than the average were flagged. The spatial average was calculated again by excluding the flagged grid points. SLP values smaller than the mean were flagged again. This way, the maximum pressure center within the South Atlantic domain is likely obtained, and secondary local SLPs maxima are eliminated. From the remaining grid points, those with SLPs greater than one standard deviation above the average were chosen to compute the position and intensity of the SAA core. When more than one group of separated grid points exists, we considered that closer to the SAA's climatological position (Fig. S8). It is worth noting that the climatology of the SAA (Fig. S8) was based on the annual mean SLP. The applied methodology yielded only one maximum pressure center over the entire domain for all calendar months.

The intensity of the SAA is defined as the areal average of the SLP over the remaining grid points. Likewise, we compute the SAA's position (i.e., longitude, latitude). Multiple centers were observed only during 19 months out of 516 months. In general, the annual cycle of SLP reveals the SAA center in its northernmost and westernmost position during austral winter (i.e., Jun-Aug; Fig. S8). The SLP in the core of SAA varies seasonally from about 1020 hPa in February to about 1024.6 hPa in August.

We estimate the trend ($\alpha$) in the SLP and the SAA position by using the least square linear regression method. The variability is estimated by the standard deviation of yearly mean values ($\sigma_{yr}$) after subtracting the long-term linear trend. The importance of the long-term historical changes relative to background climate fluctuations is estimated by the signal-to-noise ratio (S/N), which is computed as follows:

$$S/N = \frac{\Delta}{2\sigma_{yr}} = \frac{\alpha T}{2\sigma_{yr}},$$ (9)

where T stands for the time span of the ERA5 data set (i.e., 43 year) and $\Delta$ shows the changes associated with the long-term trend. In this way, a high S/N means a robust long-term change relative to the background noises (Bordbar et al., 2015, 2019). A small S/N indicates low confidence in the long-term trend.

To identify the connection between different quantities, we computed their linear correlation. To suppress the possibly misleading impacts of the seasonal cycle on the results, we subtracted the climatological monthly mean before calculating the correlation (Reintges et al., 2020).

**3 Results and discussions**

The long-term average of $W_{coast}$, $W_{curl}$, and $W_{total}$ (i.e., potential upwelling quantities), along with the spatial pattern of the surface wind and WSCD upwelling means, are shown in Fig. 1. The long-term average of $W_{coast}$ is positive (i.e., upwelling favorable) over the entire BUS. It features two pronounced peaks off the mouth of the Kunene river (~17.5°S) and Lüderitz (26.5°S). They are attributed to the local intensification of alongshore winds associated with the northward atmospheric low-level jet (Patricola and Chang, 2017). Almost over the entire domain, the long-term

mean of $W_{curl}$ is positive. The exception is a sector near the Kunene upwelling cell and corresponds to local maxima in $W_{coast}$ (~19.5°S to 17.5°S), which features downward transport (i.e., downwelling). This finding is consistent with previous studies (Fennel 1999; Bordbar et al., 2021). Off Walvis Bay, the mean of $W_{curl}$ reaches its maximum. From north of Lüderitz (~25.5°S) to south of Cape Frio (~18.5°S), the long-term average of $W_{curl}$ exceeds that for $W_{coast}$. It is reminded that $W_{curl}$ is proportional to the contrast of meridional wind stress offshore and that at the coast, i.e., $\tau^y(x)$-

$\tau^y(x=0)$. Since the spatial pattern of the offshore wind over the nBUS is fairly homogenous (Fig. 1b), the latitudinal variability of the $W_{curl}$ across the nBUS can be largely attributed to the variability of the alongshore wind at the coast. The long-term mean $W_{total}$ remains relatively invariant with latitude across the nBUS. In contrast, it is reduced southward in the sBUS (i.e., from Lüderitz to Cape Columbine). It is worth noting that in the sBUS surface wind undergoes seasonal reversal and upwelling is seasonal (Shannon 1985). The spatial pattern of WSCD upwelling

features salient features of the BUS, such as equatorward widening of the BUS, and large WSCD upwelling off Walvis Bay, which is consistent with previous studies (Fennel 1999; Bordbar et al., 2021).

The monthly anomaly correlation coefficients of the SAA (i.e., intensity, longitude, and latitude) with $W_{coast}$, $W_{curl}$, $W_{total}$, and the WSCD upwelling are shown in Fig. 2. Note that the climatological monthly mean was subtracted in each time series before computing the correlation. The anomaly correlation of the SAA intensity with $W_{coast}$, $W_{curl}$, and $W_{total}$

(Fig. 2a) reveals positive values, implying the intensified SAA is likely associated with enhanced upwelling across the entire BUS. The correlation for $W_{total}$ exceeds 0.4 almost for all latitudes, which is more robust over the sBUS (green line in Fig. 2a). Our analysis indicates that robust changes in the intensity of the SAA, in general, affect the WSCD upwelling more than the alongshore wind-driven upwelling close to the coast. The anomaly correlation of $W_{coast}$ with the SAA intensity is mostly weak (brown line in Fig. 2a). Within the most intense coastal upwelling cells in the BUS,

the Kunene and Lüderitz upwelling cell, $W_{coast}$ weakly correlates with the SAA intensity and the anomaly correlation with SAA position even vanishes. Almost over the entire domain, the anomaly correlation of $W_{curl}$ with the SSA intensity (blue line in Fig. 2a) exceeds that for $W_{coast}$. The correlation coefficient for $W_{curl}$ exceeds 0.4 from south of Cape Frio (~19°S) to north of Lüderitz (~25.5) and almost over the entire sBUS, whereas the correlation for $W_{coast}$ hardly exceeds 0.4 throughout the BUS.

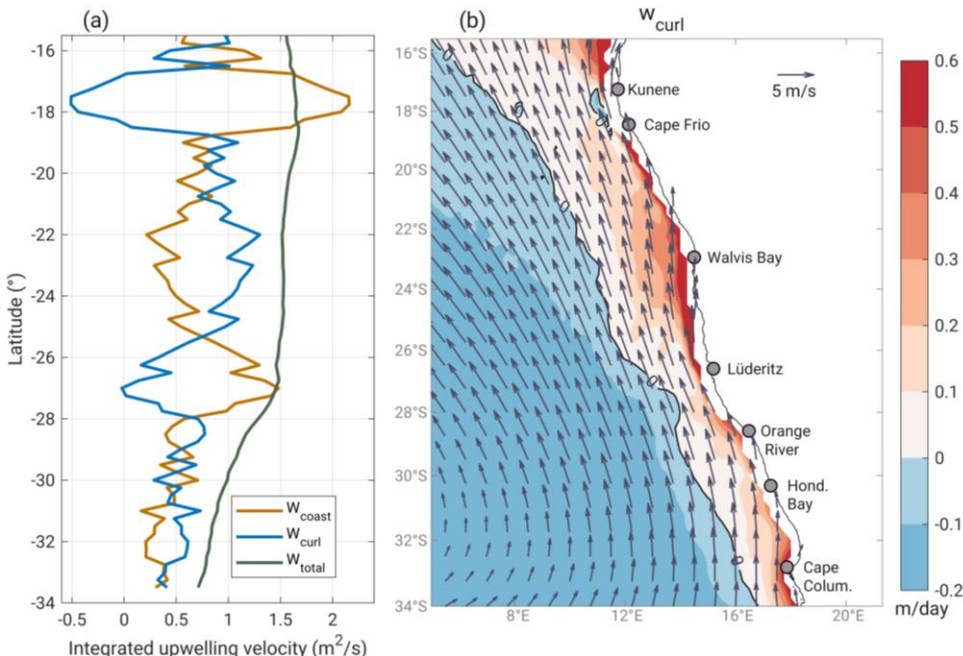

**Figure 1: Long-term average of $W_{coast}$, $W_{curl}$, $W_{total}$ (a; m$^2$/s), and spatial pattern of long-term average surface wind (b; arrows) and WSCD upwelling (b; color shaded; m/d; see equation 3) across the BUS obtained from the ERA5 wind over 1979-2021. Contours in panel b indicates where the WSCD upwelling is equal to zero.**

The anomaly correlation of potential upwelling quantities (i.e., $W_{total}$, $W_{coast}$, $W_{curl}$) with the SAA longitude is weak (Fig. 2b). With regard to the SAA latitude (Fig. 2c), the correlation coefficient for $W_{curl}$ and $W_{total}$ is negative and smaller than -0.4 over the entire sBUS, meaning southward displacement of the SAA is associated with an enhanced $W_{curl}$ and $W_{total}$. However, the correlation for $W_{coast}$ is relatively weak. For the nBUS, these correlations are not statistically significant.

The anomaly correlation between the SAA intensity and the WSCD upwelling (Fig. 2d) is broadly positive across the entire BUS, meaning an intensification of SAA is likely associated with a strengthening of WSCD upwelling. Indeed, the spatial pattern of the correlation between the SAA intensity and the WSCD upwelling is reminiscent of the long-term average of the WSCD upwelling (Fig. 1b), with an enhanced value off Walvis Bay and south of Lüderitz. This is consistent with the correlation between SAA intensity and $W_{curl}$ (Fig. 2a). The WSCD upwelling anomaly in the sBUS appears to be significantly affected by the meridional displacement of the SAA (Fig. 2f). Generally, the SAA poleward excursion is likely resulting in an enhanced WSCD upwelling over the entire sBUS. In the whole BUS, the anomaly correlation between the WSCD upwelling and the SAA longitude is weak (Fig. 2e), which is consistent with the correlation between $W_{curl}$ and the SAA longitude (Fig. 2b). Overall, these anomaly correlation patterns indicate that any of the SAA systematic changes have different consequences for the wind-driven upwelling in the nBUS and sBUS.

We ask for specific correlation patterns for the austral summer or winter months, corresponding to maximum or minimum solar radiation. Repeating the same analysis for the austral winter (Jun-Aug; Fig. S9) and summer (Dec-Feb; Fig. S10), the general structures of the anomaly correlations do not change much. Overall, the size of correlations is higher in the austral winter than in the austral summer. Wintertime $W_{total}$ and the SAA intensity are closely connected, with the correlation coefficient is higher than 0.5 for most of the BUS. In sBUS, wintertime $W_{total}$ and the SAA latitude

are strongly anti-correlated, with a correlation coefficient smaller than -0.6. It suggests that the poleward excursion of the SAA in boreal winter is very likely associated with an enhanced $W_{total}$ across the sBUS. Since the coastal upwelling undergoes a strong seasonal cycle south of the Lüderitz upwelling cell (~27°S), including the seasonal cycle in the correlation would yield a different pattern.

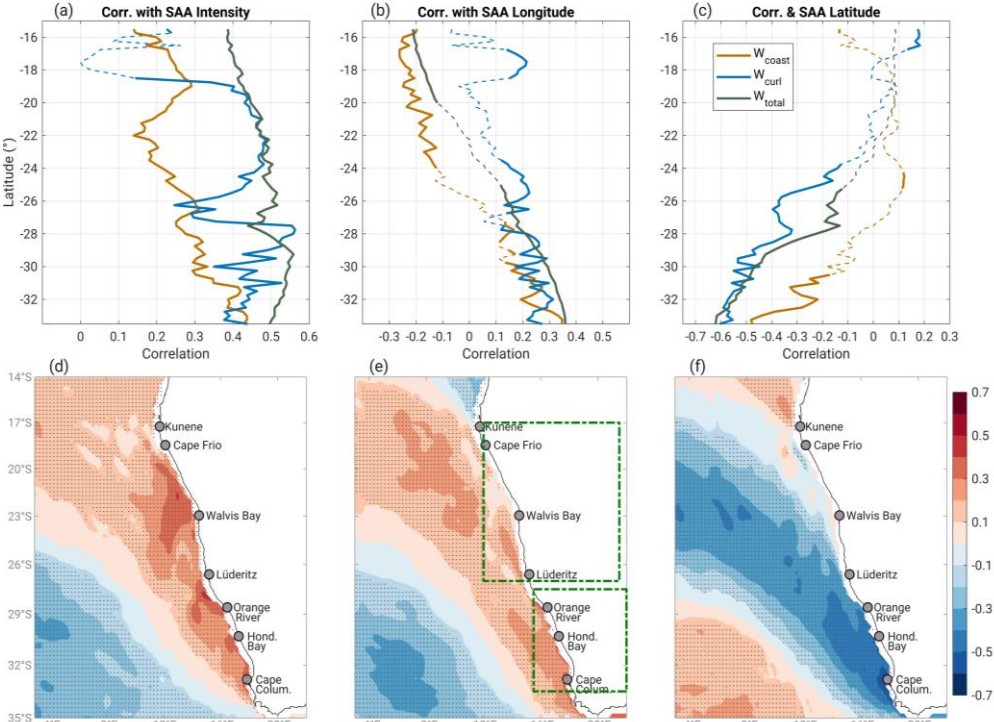

**Figure 2: Monthly anomaly correlation of the SAA intensity (a), longitude (b), and latitude (c) with $W_{coast}$ (brown line), $W_{curl}$ (blue line), *and* $W_{total}$ (green line) in each latitude across the BUS. In panels a-c, solid lines represent the correlations that are statistically significant at the 99% confidence level, whereas dash lines denote statistically insignificant correlations. The correlation of the WSCD upwelling with the intensity (d), longitude (e), and latitude (f) of the SAA are displayed in bottom panels. Stippled areas in d-f indicate where the correlation is statistically significant at the 99% confidence level.**

We found the wind-driven upwelling in the BUS is significantly affected by the variations of the SAA. We also assess the changes in the regional horizontal pressure gradient, which determines geostrophic winds and have strong influence on the local winds (Lamont et al., 2018). The differences between the SLP over the SAA core and the areal-averaged SLPs over the nBUS and sBUS (used boxes are marked in Fig. 2e) are considered as approximation for the SLP gradient related to the surface wind regimes. In general, the rate of warming over land is larger than the adjacent ocean across the BUS in 1979-2021 (Fig. S11). This feature is more pronounced in the BUS northernmost sector (i.e., Cape Frio and Kunene upwelling cells), where the rate of warming over land exceeds 0.05 °C/year. Thus, one may expect an enhanced SLP gradient between the land and the adjacent ocean.

The time series of the SLP over the SAA and its contrast with the SLP in the nBUS and sBUS show marginal positive trends of about 0.020, 0.014, and 0.015 hPa/yr, respectively (Fig. 3a). But the interannual variation of the SLP is large and reduces confidence in the significance of the trend. The S/N of the trends is typically smaller than 1.0. The time series of the SLP gradients follow the SLP of the SAA core reasonably well, implying that the variability of the

SLP gradients is largely related to the SAA. We repeated these analyses for different rectangular boxes and found that the trends of the SLP gradients were not sensitive to the size of rectangular boxes (not shown). There is a slight westward and poleward migration of the SAA of about -0.15°E, and -0.50°N degrees over the observation period of 43-year, respectively (Fig. 3b). The small S/N does not allow for a meaningful statement about the SAA long-term excursion. The time series of the SAA longitude shows a wide range of zonal SAA excursions between about 30°W and about 5°E. Several events with a large excursion of the SAA occurred every few years, i.e., on an interannual timescale, but sometimes an anomalous zonal displacement persisted for a few years. For example, the years 1997 and 2006 are characterized by persistent eastward SAA displacements, and the years 1986, 2001, 2010, and 2017 feature anomalous westward SAA migrations. The meridional position of the SAA ranges from 35°S to 25°S.

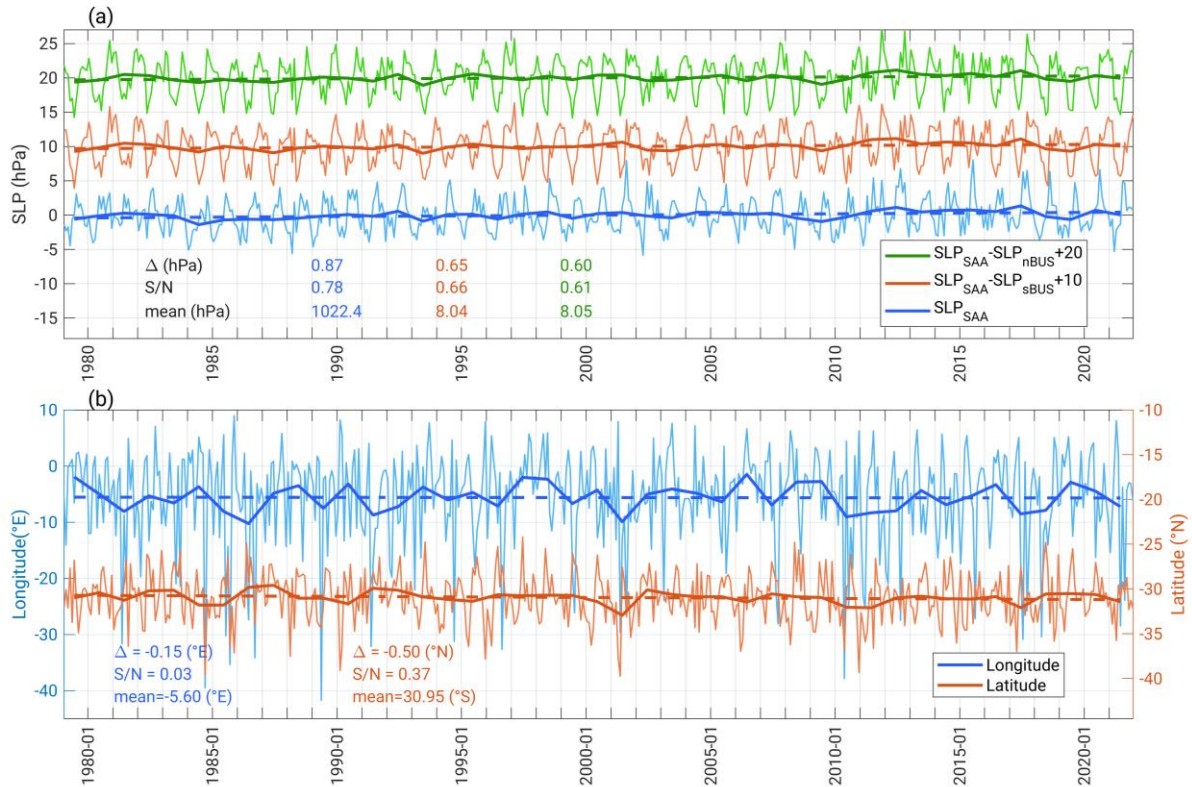

**Figure 3: a, Time series of the SLP in the SAA core (blue lines) and its difference with the SLP over nBUS (green lines) and sBUS (red lines). The SLP for the nBUS and the sBUS is the average over the rectangular boxes displayed in Fig. 2 e. The long-term mean was subtracted from each time series. Please note the offsets for the nBUS (+10 hPa) and sBUS (+20hPa) time series. In panel (b), the time series of longitude (blue; °E) and latitude (red; °N) of the SAA core are shown. In both panels, thick and thin solid lines denote the yearly and the monthly means. Dashed lines represent the trend lines fitted to the yearly mean values. At the bottom of each panel, the long-term changes associated with trend line (i.e., Δ), the corresponding S/N, and the mean are shown by identical colors as the time series.**

We computed the SAA intensity and position time series for Jan-Feb and Jun-Aug, corresponding to austral summer and winter, respectively (Fig. S12). Again, a positive trend in the summertime and wintertime SAA intensity is found, but the corresponding S/N remains smaller than unity. Despite the accelerated rate of global average temperature over 1990s (Bordbar et al., 2019), the SAA intensity and position remained steady and underwent no significant trend. It is

consistent with the previous study (Polonsky & Serebrennikov, 2021), which reported a hiatus in the intensification of the coastal upwelling across the BUS since the 1990s. If there is any tendency in the intensity and location of the SAA due to global warming, it is presumably too small to emerge from background climate fluctuations.

To investigate this further, the long-term trend in the yearly, Jul-Oct, and Jan-Apr SLP means are shown in Fig. 4. The SLP trend appears to be positive almost over the entire South Atlantic. However, the pattern of rising SLP is not uniform, and the maximum trend is not found in the SAA center. In general, the size of the trend varies more in the meridional direction and is more prominent over higher latitudes, particularly for July-October. The most prominent trend is found in the southwest and the southeast of the domain in Jul-Oct (Fig. 4c) and Jan-Apr (Fig. 4b), respectively.

The structure of the trend reminds the recent multi-decadal trend in the SAM, which is associated with an enhanced meridional SLP gradient between the polar and mid-latitudes (Wachter et al., 2020; Fogt & Marshall, 2020).

The SLP trend over the center of SAA is about 0.02 hPa/yr (i.e., $\Delta$ of ~0.86 hPa) and larger than that near the coast of Namibia and South Africa. It indicates a slight enhancement of the SLP gradients between the SAA and the BUS coastal zones. This enhanced SLP gradient appears to be slightly more (less) pronounced for the Jul-Oct (Jan-Apr)

historical changes (Fig. 4b, c).

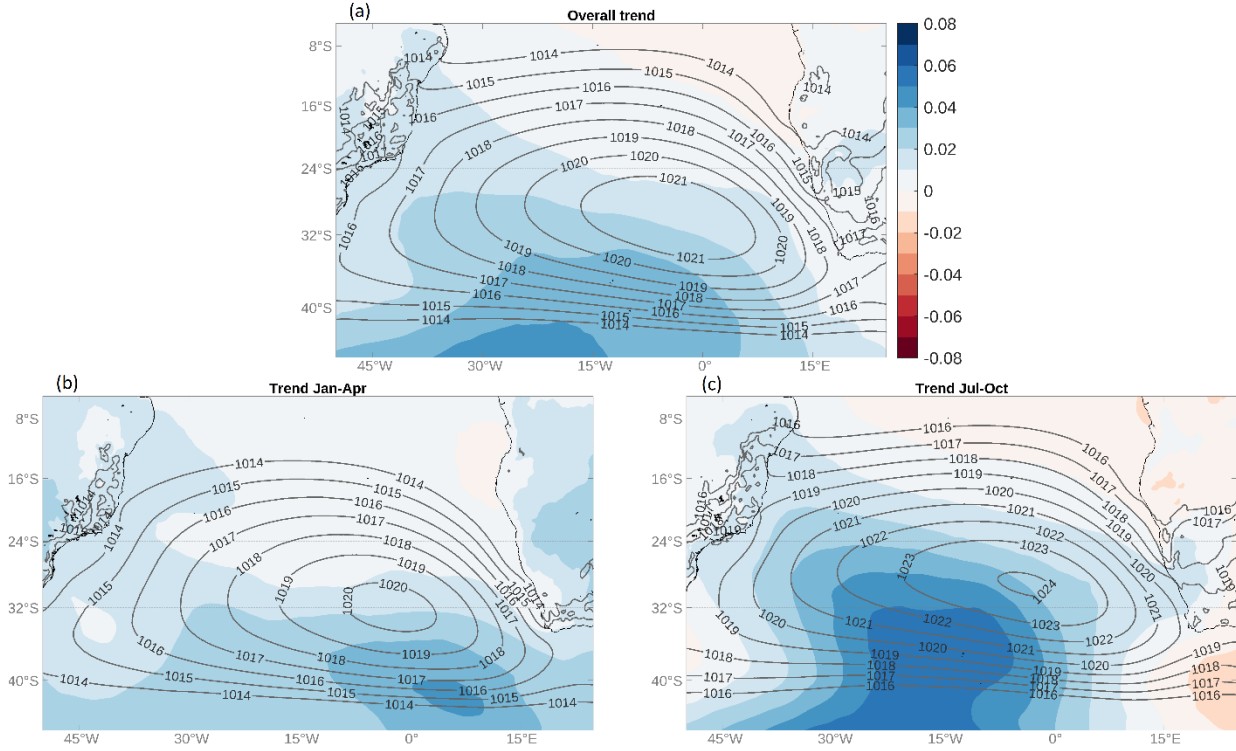

**Figure 4: Long-term mean (contours; hPa) and trend (color shading; hPa/yr) of the ERA5 SLP for 1979-2021. Panel (a) represents the trend ($\alpha$) and mean-state obtained for yearly mean SLP, whereas (b-c) corresponds to the averages over Jan-Apr and Jul-Oct, respectively.**

In Fig. 5, the variability (i.e., $\sigma_{yr}$) of the SLP and the S/N of the long-term SLP trends are displayed. The S/N indicates whether the long-term trend can emerge from the background climate fluctuations. The level of the SLP fluctuations is enhanced poleward across the entire domain (contours in Fig. 5). The SLP variability in the southern sector is more than twice that in the north and central parts. The year-to-year variations are considerable for the

wintertime when the region's meridional SST gradient reaches its seasonal maximum. Further, severe cyclones, anticyclones, and frontal passages are more frequent during wintertime. For the yearly mean SLP (Fig. 5a), the S/N barely exceeds 1. An exception is an area between 40°-35°S and 15°W-5°W with S/N of about 1.3. Small S/N highlights that the historical trends do not come to light in the presence of strong climate variability. For both, winter- and summertime historical trends the S/N remains smaller than one over almost the entire domain, (Fig. 5b,c). Hence, the long-term SLP trends should be interpreted with caution. The time series of the SAA intensity and its geographic position further supports this result (Fig. 3, Fig. S12).

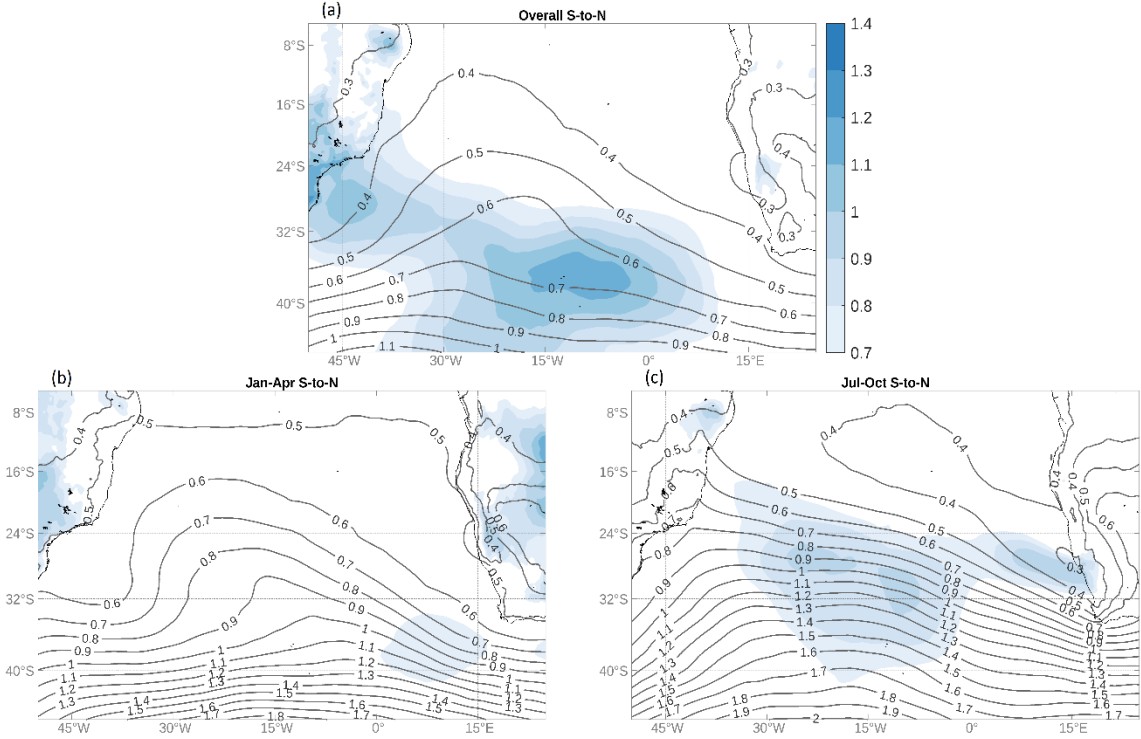

**Figure 5: Spatial pattern of the variability (i.e., $\sigma_{yr}$; contours; hPa) and signal-to-noise ratio (S/N) associated with the long-term trend (color shading) of the ERA5 SLP over 1979-2021. Panel (a) represents the values obtained for yearly mean SLP, whereas panels (b) and (c) correspond to the average over Jan-Apr and Jul-Oct, respectively.**

In the following, we further assess the historical changes of potential upwelling quantities, including $W_{coast}$ and $W_{curl}$, in Kunene, Walvis Bay, Lüderitz, and Cape Columbine upwelling cells (Fig. 6-8). In the Kunene cell, positive $W_{coast}$ (i.e., upwelling favorable) persists throughout the year (Fig. 6a, 8a). The annual mean $W_{coast}$ is around 1.96 m²/s. The related maximum upwelling velocity (i.e., x=0) approximated by using equation 5 (see methods) is roughly 6.9 m/d, which is one order of magnitude larger than the typical size of WSCD upwelling (i.e., $w_{curl}$) across the BUS (see Fig. 1b). The distribution of $W_{coast}$ in Kunene cell is skewed towards strong upward transport and exhibits interannual variability. In addition, the number of severe $W_{coast}$ indicated by outliers remains nearly steady over the considered period. Compared with that of $W_{coast}$, the distribution of $W_{curl}$ in Kunene cell has smaller skewness (Fig. 7a). Unlike $W_{coast}$, $W_{curl}$ is not perennial. It also undergoes a strong interannual variability. The number of days with positive $W_{curl}$ in a year varies from 110 to 160 (Fig. 8b). Outliers in $W_{curl}$ appeared almost every year.

Off Walvis Bay, the mean of $W_{coast}$ is about 0.35 m²/s. Based on equation 5, upwelling velocity is roughly 2.0 m/d.
Negative $W_{coast}$ (i.e., downwelling favorable) appeared more frequently off Walvis Bay than in the Kunene upwelling
cell (Fig. 6b). However, the number of days with positive $W_{coast}$ off Walvis Bay remains higher than 300 per year (Fig.
8c). Since the late 1990s, there has been a slight trend toward stronger $W_{coast}$, with several vigorous upwelling events
that can be considered as anomalous events. A large interannual variation in $W_{coast}$ off Walvis Bay is visible. Persisting
throughout the year, positive $W_{curl}$ off Walvis Bay is the strongest among the upwelling cells (Fig. 7b, 8d). The number
of days with the upwelling-favorable $W_{curl}$ is higher than 340 per year in the Walvis Bay upwelling cell. This number
remained steady with no significant trend.

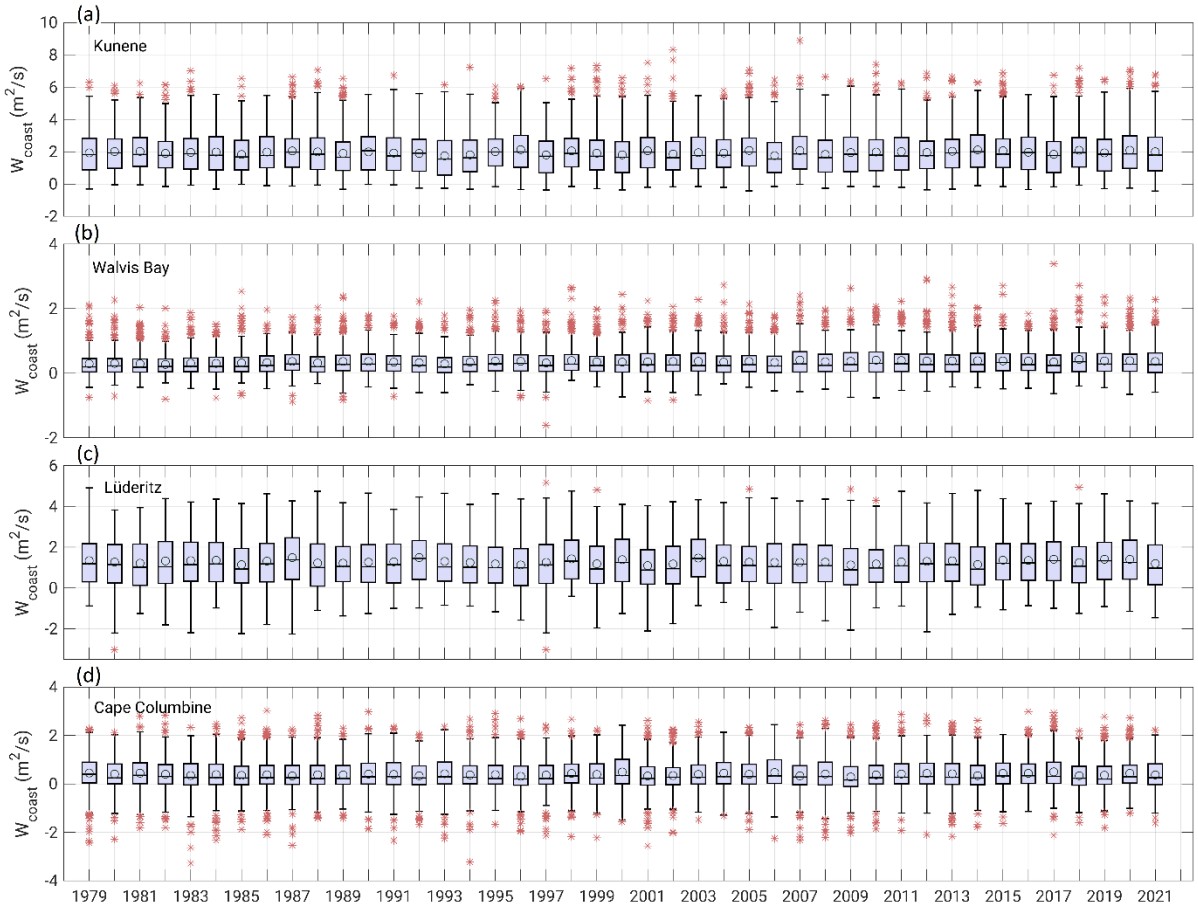

**Figure 6: Boxplot representing the median and the interquartile range of $W_{coast}$ in Kunene (a), Walvis Bay (b), Lüderitz (c),**
**and Cape Columbine (d) upwelling cells derived from ERA5 data over 1979-2021. Each box represents the distribution of**
**the daily $W_{coast}$ in the corresponding year. The bands and circles inside the boxes represent the medians and mean,**
**respectively. The red crosses indicate the extreme events defined as the values exceeding the confidence limits (i.e., outliers).**
**$W_{coast}$ for the Kunene, Walvis Bay, Lüderitz, and Cape Columbine are the average over 17.5°S-17°S, 23.5°S-23°S, 27°S-**
**26.5°S, and 33.25°S-32.75°S, respectively.**

Within the Lüderitz cell (Fig. 6c, 8e), positive $W_{coast}$ is almost year-round, with a long-term mean of about 1.28
m²/s. Based on equation 5, this corresponds to a maximum upwelling velocity (i.e., x=0) of about 6.9 m/d. The positive
$W_{coast}$ occurred at more than 300 days in a year (Fig. 6c, 8e). However, there are also episodes with negative $W_{coast}$.

Only a few outliers in $W_{coast}$ off Lüderitz (e.g., 2005, 2009, 2010, 2018) were observed. In contrast, outliers in $W_{curl}$ off Lüderitz are observed almost every year (Fig. 7c). The number of days with positive $W_{curl}$ fluctuates around 220 per year (Fig. 8f).

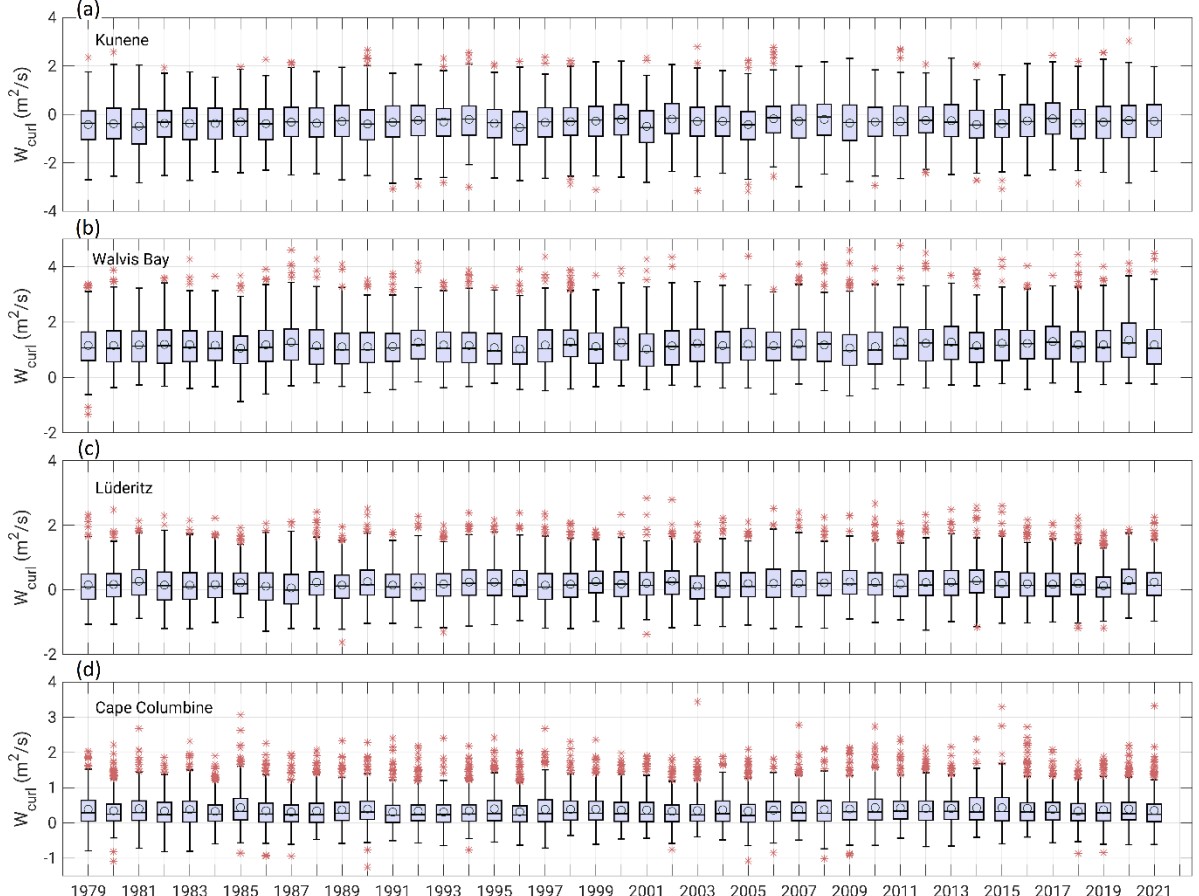

**Figure 7: Boxplot representing the median and the interquartile range of $W_{curl}$ in Kunene (a), Walvis Bay (b), Lüderitz (c), and Cape Columbine (d) upwelling cells derived from ERA5 data over 1979-2021. Each box represents the distribution of the daily $W_{curl}$ in the corresponding year. The bands and circles inside the boxes represent the medians and mean, respectively. The red crosses indicate the extreme events defined as the values exceeding the confidence limits (i.e., outliers). $W_{curl}$ for the Kunene, Walvis Bay, Lüderitz, and Cape Columbine are the average over 17.5°S-17°S, 23.5°S-23°S, 27°S-26.5°S, and 33.25°S-32.75°S, respectively.**

Off Cape Columbine, the seasonal reversal in $W_{coast}$ is more pronounced than in other upwelling cells (Fig. 6d). The long-term mean of $W_{coast}$ is about 0.39 m²/s. Based on equation 5, the maximum upwelling velocity is about 2.5 m/d. The number of days with positive $W_{coast}$ is around 270 per year (Fig. 8g). $W_{coast}$ does not show a long-term trend but undergoes interannual variability. As for $W_{curl}$ (Fig. 7d), outliers are primarily positive. The days with positive $W_{curl}$ fluctuate around 300 per year (Fig. 8h).

We also apply the Mann-Kendall statistical test to examine the significance level of the trend in the time series of yearly mean $W_{coast}$ and $W_{curl}$ in the selected upwelling cells. There is no statistically significant trend at the 99% confidence interval in almost all upwelling cells. The exception is the slight positive trend in the yearly mean $W_{coast}$ off Walvis Bay. One should keep in mind that the time series are not sufficiently long to identify the potential impacts of

modes of climate variability, such as the SAM, the Interdecadal Pacific Oscillation, and Atlantic Multidecadal Variability, which all span variability on decadal timescales and beyond.

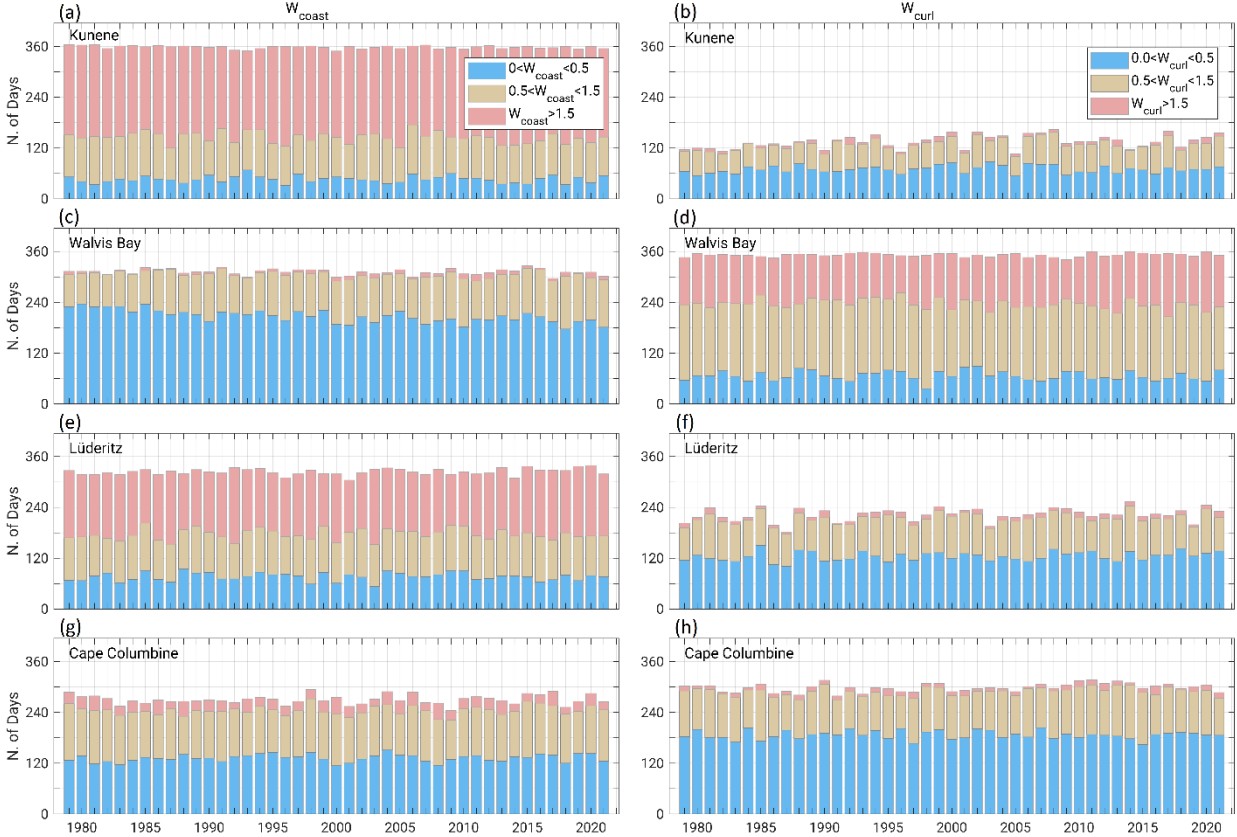

Figure 8: Number of days per year with upwelling favorable $W_{coast}$ (a,c,e,g) and $W_{curl}$ (b,d,f,h) in Kunene (a,b), Walvis Bay (c,d), Lüderitz (e,f), and Cape Columbine (g,h). Light blue, light yellow, and light red bars indicate weak ($0.0 < W < 0.5$ $m^2/s$), moderate ($0.5 < W < 1.5$ $m^2/s$), and strong ($W > 1.5$ $m^2/s$) transport.

**4 Summary**

To get new insight into the multi-decadal variation of the South Atlantic Anticyclone (SAA) and its link to the wind-driven upwelling across the Benguela Upwelling System (BUS), the observed sea level pressure (SLP) and surface wind datasets from ERA5 archive were analyzed.

We introduced the concept of potential upwelling which indicates upwelling processes driven solely by the vertical flux of atmospheric horizontal momentum into the ocean. This approach helps to focus on well-defined quantities, even though they do not reflect a comprehensive picture of processes that can considerably modify the upwelling (i.e., alongshore wind variability, coastally trapped waves, frontal dynamics, etc.). We consider three potential upwelling quantities, which are the cross-shore integral of wind-driven coastal upwelling ($W_{coast}$), the wind-stress-curl-driven upwelling ($W_{curl}$), and their summation ($W_{total}$). The detail structure of the wind over the coastal zones plays a significant role in local wind-driven coastal upwelling and wind-stress-curl-driven upwelling. However, we show that the detailed wind pattern over the coastal zone does not play a significant role for $W_{total}$. Since it provides an accurate estimate of $W_{total}$, this approach is promising and suitable for the ERA5 data, even though the spatial resolution of this data is too

coarse (i.e., 0.25°×0.25°) to resolve the small-scale processes driven by coastal SST fronts, orographic features near the coast, which can alter the structure of surface winds and amplify or reduce the coastal wind drop-off.

We found a robust anomaly correlation between the SAA intensity, $W_{total}$, and $W_{curl}$ over almost the entire BUS. However, this correlation for $W_{coast}$ is much weaker. One possible explanation is that according to our definition in this study, the $W_{coast}$ across the BUS depends solely on the meridional wind at the coast, which is significantly influenced by local processes. In contrast, variation of the $W_{total}$ is solely driven by the fluctuations of meridional winds offshore, and $W_{curl}$ is related to the meridional wind stress both offshore and at the coast. Given the fact that the time variation of offshore wind is closely related to the SLP anomalies in the SAA, one expects a closer connection of $W_{total}$ and $W_{curl}$ to the SAA relative to that for the $W_{coast}$.

In general, an intensified SAA is likely accompanied by an enhanced $W_{total}$ and $W_{curl}$. For $W_{total}$, this connection seems to be more pronounced in the poleward sector of the BUS. For $W_{curl}$, the relationship with the SAA intensity is most robust off Walvis Bay and south of the Lüderitz upwelling cell, which feature a larger $W_{curl}$ long-term mean relative to the rest of the BUS. Further, a southward SAA excursion is likely associated with strengthening $W_{total}$ and $W_{curl}$ over the poleward portion of the BUS. This connection is more pronounced during austral winter (i.e., June-August). Our findings suggest that robust changes in the SAA (i.e., intensity and position) affect the wind-stress-curl-driven upwelling more than the alongshore wind-driven coastal upwelling in the BUS. Thus, any systematic changes in the SAA can have different implications across the Benguela ecosystems.

Despite a slight upward SLP trend in the subtropical South Atlantic during 1979-2021, the ratio between changes associated with the long-term SLP trend (i.e., Δ) and the standard deviation of the long-term trend subtracted yearly mean SLP is small across the entire domain. Further, potential upwelling quantities, including $W_{coast}$ and $W_{curl}$, in several upwelling cells remained steady. Generally, they exhibited neither a statistically significant long-term trend nor prominent changes in the characteristics of the variability (i.e., median, interquartile ranges, and extremes). Overall, our results neither demonstrate nor rule out the potential impacts of anthropogenic global warming on the atmospheric drivers of upwelling in the BUS. A possible explanation is that a much longer time is likely required to detect the robust global warming signals in the wind-driven upwelling across the BUS.

The analyzed ERA5 data sets are presumably too short for detecting the regional wind-driven upwelling systematic changes associated with global warming. Hence, it remains difficult to examine Bakun's hypothesis reliably. Indeed, one cannot attribute the entire wind field variability over the BUS solely to the SAA and the regional cross-shore surface air temperature gradient. Localized drivers of the surface winds (e.g., sub-mesoscale SST fronts, orographic effects, eddies, and land-sea breezes), which the operational model used in ERA5 reanalysis does not resolve, may significantly alter the surface wind field. In addition, short- and long-term fluctuations in the strength and position of the coastal wind drop-off, which are of great importance for the regional marine ecosystem studies, cannot be reliably addressed by ERA5 data.

Furthermore, one should not neglect the internally and externally forced variations of the key components of the general atmospheric circulation (i.e., the Hadley Cell, the Walker Circulation) and their potential impacts on the SAA and the surface winds across the BUS (Gillett et al., 2006; Gilliland & Keim, 2017; Rouault & Tomety, 2022).

It is worth noting that the enhanced ocean stratification is a well-established consequence of ocean warming, suppressing the upwelling efficacy in nutrient supply. Thus, an intensified upwelling-favorable wind does not necessarily imply an enhanced nutrient availability in the euphotic zones.

The analysis conducted in this study is broadly transferrable to other major Eastern Boundary Upwelling Systems and will be beneficial for regional marine ecosystem studies.

*Data availability.* ERA5 SLP and surface wind (Hersbach et al., 2018) were obtained from https://cds.climate.copernicus.eu. ASCAT surface winds (Ricciardulli and Wentz, 2016) were downloaded from https://remss.com/missions/ascat/. Observed SLP in the St. Helena Island (Feistel et al., 2003) can be publically
accessed at https://www.io-warnemuende.de/hix-st-helena-island-climate-index.html.

*Author contributions.* MHB, MSCH, and VM developed the methodology and contributed to interpreting the results, discussion, and refinement of the paper. MHB wrote the first draft of the paper.

*Conflicts of Interests.* The authors declare no conflict of interest.

*Acknowledgements.* This study was conducted within the frame of the EVAR and BANINO project sponsored by the
BMBF with the reference number 03F0814, and 03F0795B, respectively. This study has been conducted using E.U. Copernicus Marine Service Information. We thank Dr. Fabien Desbiolles and the anonymous reviewer for their constructive comments.

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
