# Peer review of "Low confidence in multi-decadal trends of wind-driven upwelling across the Benguela Upwelling System"

_Earth System Dynamics, 2023_

## Author Comment (AC1)

We thank the anonymous reviewer for his/her constructive comments. We overall agree with the points raised, which have been considered in revising the manuscript. In the following, our responses to the reviewers are shown in *blue italics*.

**Reviewer #1**

The present study analyses the coastal upwelling regime at the eastern boundary of the South Atlantic. The focus is on multi-decadal variability of the wind forced total upwelling composed of coastal upwelling due to alongshore winds and wind-stress-curl-driven upwelling. The manuscript analyses a specific reanalysis dataset of the global climate, ERA5. This dataset is somehow validated against few other available datasets. Total upwelling and wind-curl-driven upwelling is found to be strongly related to the strength and position of the South Atlantic Anticyclone (SAA). Long-term trends identified in the ERA5 dataset are mostly weak, showing considerable heterogeneity, and have only a small signal-to-noise ratio due to enhanced interannual climate variability. My major concerns are:

1) ERA5 is a reanalysis dataset that includes information about uncertainties for all variables. I think it would be necessary to evaluate this uncertainty with regard to possible trends. Particularly trends in the wind forcing are still highly uncertain and a clear description what are the uncertainties provided with the dataset is necessary.

*__Response:__ We agree with the point raised by the reviewer about the data uncertainty in ERA5 reanalysis. We addressed this comment as follows:*

- *We have now compared the monthly, yearly, and 11-year running mean of ERA5 SLP with the observed SLP in St. Helena Island (see Figure S1). In addition, we computed the long-term SLP trend in both SLPs, which are very small and at about -0.002 and 0.005 hPa/yr for the ERA5 and observation.*

- *We computed the ERA5 and ASCAT meridional wind trend, the main contributor to alongshore wind-driven upwelling and WSCD upwelling. Results are shown in Fig. S6.*

- *The observation-based data in the BUS is very sparse in time and space. Finding continuous and reliable records is a challenge. We found four WMO climate stations in Angola, Namibia, and South Africa, including Luanda, Benguela, Port Nolloth, and Cape Town, all located near the coast and typically a few 10 meters above the sea surface. We have now compared the monthly mean surface pressure from these stations and the SLP from the ERA5. The results are shown in Fig S2.*

*We have now discussed above mentioned points in lines 155-178 "To examine the accuracy of ERA5 data over the BUS, we utilize several observations, including satellite-derived daily ASCAT surface winds covering 2007-2021 (Ricciardulli and Wentz, 2016), in situ SLP measurements in St. Helena Island (5.7°W-15.95°S) with a long-term record from 1893 to the present (Feistel et al., 2003). We compare the observed and simulated fluctuations on monthly to decadal time scales. We also compare the long-term trends in the observations and the ERA5 reanalysis. Further, we qualitatively examine the ERA5 SLP by using the time series of observed surface pressure in several climate stations near the southwest African coast, including Luanda (13.25°E, 8.85°S), Benguela (13.42°E,12.58°S), Port Nolloth (16.87°E,29.23°S), and Cape Town (18.60°E,33.96°S). For a given weather regime, the SLP in a station represents the surface air pressure if the station was located at the altitude of global mean sea level. The elevation of the selected climate stations from the global mean sea level is typically less than 80 meters; therefore, it is reasonable to assume that the observed surface pressure closely follows the SLP time series.*

*Consistent with a previous study (Belmonte Rivas and Stoffelen 2019), ERA5 for the southwest African coast agrees well with available observation-based data sets (Supplementary Info; Fig. S1-6). For St. Helena Island, the time series of the monthly, yearly, and 11-year running SLP means obtained from ERA5 evolved very closely with those derived from the observation (Fig. S1). The time series of simulated and observed yearly mean SLPs display only marginal trends, which are about 0.0025 and -0.0056 hPa/year, respectively. The available time series of surface pressure observed at Luanda, Benguela, Port Nolloth, and Cape Town climate stations evolve closely with the EAR5 SLP (Fig. S2). However, the surface pressure time series for all stations is not continuous or sometimes appear to have offset accuracy issues (e.g., Benguela; see Fig. S2b), which do not allow the evaluation of long-term trends. The spatial pattern of the ERA5 and ASCAT meridional wind trends for 2008-2021 agree reasonably well (Fig. S6). Overall, meridional wind near the coastal area of the BUS underwent an upward trend over the last decade. The spatial structure of the trend in the model appears to be smooth relative to that derived from ASCAT wind. In addition, the wind intensification south of 30°S is pronounced in the ASCAT winds, whereas a relatively small trend is derived from ERA5 data."*

*".*

2) There is a validation of the ERA5 dataset with respect to the satellite-derived ASCAT wind dataset and to sea level pressure in St. Helena. This validation focuses on short-term or seasonal variability. However, to strengthen to topic of the paper it would be important to look at the comparison of multi-decadal variability. How good do ERA5 represent long-term changes identified in ASCAT or St. Helena sea level pressure?

**Response:** *This is a valid point. We computed the long-term trend for both data sets and the 11-year moving average for the St. Helena SLPs (please see Fig. S1 and S6). We have now discussed the results in the main text. Please see lines 167-179 "For St. Helena Island, the time series of the monthly, yearly, and 11-year running SLP means obtained from ERA5 evolved very closely with those derived from the observation (Fig. S1). The time series of simulated and observed yearly mean SLPs display only marginal trends, which are about 0.0025 and -0.0056 hPa/year, respectively. The available time series of surface pressure observed at Luanda, Benguela, Port Nolloth, and Cape Town climate stations evolve closely with the EAR5 SLP (Fig. S2). However, the surface pressure time series for all stations is not continuous or sometimes appear to have offset accuracy issues (e.g., Benguela; see Fig. S2b), which do not allow the evaluation of long-term trends. The spatial pattern of the ERA5 and ASCAT meridional wind trends for 2008-2021 agree reasonably well (Fig. S6). Overall, meridional wind near the coastal area of the BUS underwent an upward trend over the last decade. The spatial structure of the trend in the model appears to be smooth relative to that derived from ASCAT wind. In addition, the wind intensification south of 30°S is pronounced in the ASCAT winds, whereas a relatively small trend is derived from ERA5 data".*

3) Inherent to the discussion in the manuscript is that long-term changes are associated with human-induced global warming and shorter-term (interannual to decadal) variability to internal variations of the climate system. This is an assumption that cannot be proven with the current dataset. I would suggest to more carefully discuss multi-decadal, decadal and interannual variability pointing to possible mechanisms that could be associated with internal climate variability or global warming.

**Response:** *We agree that our study neither proves nor denies the potential influences of anthropogenic global warming. We have carefully revisited our manuscript and avoided any*

*definite statement about the changes due to internal variability or anthropogenic global warming. For example, please see lines 378-379 "If there is any tendency in the intensity and location of the SAA due to global warming, it is presumably too small to emerge from background climate fluctuations" and lines 490-494 "Overall, our results neither demonstrate nor rule out the potential impacts of anthropogenic global warming on the atmospheric drivers of upwelling in the BUS. A possible explanation is that a much longer time is likely required to detect the robust global warming signals in the wind-driven upwelling across the BUS".*

*We have now change the title of our manuscript to "Low confidence in multi-decadal trends of wind-driven upwelling across the Benguela Upwelling System".*

This is an overall well-written manuscript that contribute to our understanding of the wind-driven upwelling, its variability and its long-term changes. It might be acceptable after revising the raised major and specific point.

Specific points, comments and suggestions:

Abstract: In the abstract should be mentioned that the study is based on ERA5 reanalysis.

*__Response:__ We have mentioned ERA-5 in the abstract. Please see lines 9-10 "Using the European Centre for Medium-Range Weather Forecasts ERA5 reanalysis for 1979-2021, …".*

L11: "cross-shore integral of wind-driven coastal upwelling": What is this? Do you mean coastal upwelling driven by alongshore winds?  The different upwelling terms must be clearly defined and understandable. Also L15: "integrated wind stress curl-driven and total upwelling": again term is not well defined and there should be a simpler way to introduce the three terms: coastal upwelling driven by alongshore winds, wind stress curl driven upwelling in the near coastal band and total upwelling as the sum of both.

*__Response:__ We have now rephrased the sentence as the reviewer recommended. Please see lines 10-12 "we investigate multi-decadal changes of the South Atlantic Anticyclone and their impacts on coastal upwelling driven by alongshore winds, wind-stress-curl-driven upwelling closed to the coastal band and total upwelling as the sum of both across the Benguela Upwelling System".*

L16: "more signatures" is unclear

*Response:* *As the reviewer suggested, we have deleted this part of the sentence and added a new sentence to make the abstract more understandable. Please see lines 17-19 "The upwelling in the equatorward portion of the Benguela Upwelling System is significantly affected by the anticyclone intensity. In contrast, the poleward portion is also influenced by the meridional position of the anticyclone. In general, the impacts of the anticyclone on the local upwelling are more robust during the austral winter".*

L20: Sentence "However, …" implies that internal climate variability exists only on shorter timescales and that trends are necessarily due to other forcing. Would it be better to write "… can be obscured by interannual to decadal climate variability"?

*Response:* *We agreed. We have modified the sentence to (lines 20-21) "However, this trend features a small signal-to-noise ratio and can be obscured by interannual to decadal climate variability".*

L28: These modes result …

*Response:* *Done. Please see line 29 "These modes result from".*

L28: "nonlinear climate dynamics" Why nonlinear? There are many linear climate feedbacks. Maybe: coupled ocean-atmosphere dynamics.

*Response:* *That is a valid point. We have modified the sentence to (Lines 29-31) "These modes result from the interactions between different components of the Earth's climate system (i.e., atmosphere, ocean, cryosphere, etc.) and modulate the ocean-atmosphere heat, mass, and momentum exchanges on the regional or even global scale".*

L31: at the eastern margin

*Response:* *Corrected (Line 32).*

L37: "Farther offshore, …" brackets are not logical: divergence is associated with upward, convergence with downward velocity. I would suggest to remove the terms in the brackets.

*Response: We have now removed the brackets (Lines 38).*

L48: "Based on …". There is a rich literature on the local and remote forcing of Benguela Niños and Niñas. This sentence do not reflect our current understanding of these anomalous coastal warm and cold events.

*Response: We have now explained the importance of remote and local influences in the onset and development of Benguela Niños. Please see lines 50-56: "Warm events are associated with the southward intrusion of low-oxygenated and nutrient-enriched equatorial waters with severe consequences for marine ecosystems. Anomalies in the local wind stress and associated coastal upwelling along the southwest African coast and the relaxation of trade winds over the western equatorial Atlantic, which excites Kelvin waves propagating eastward along the equator and thereafter southward along the west African coast, play central roles in the onset and development of the Benguela Niños (Shanon et al., 1986; Richter et al., 2010; Lubbecke et l., 2010). These wind anomalies are often associated with slowing down in the subtropical South Atlantic Anticyclone (SAA)".*

L55: "equatorward part of the BUS" requires a definition, e.g. by a latitude range.

*Response: We have now mentioned the latitude range in lines 61-62 "From the northernmost part (~16°S) of the BUS to the north of Lüderitz (~27°S), …".*

L58: "poleward portion", same as before. Please provide a definition of the region.

*Response: We have now mentioned the latitude range of poleward portion of the BUS in line 64 "From the south of Lüderitz (~27°S) to the southernmost part of the BUS (~33°S) …".*

L59-61: At least there is some uncertainty about the role of southward migration of the SAA. The southward movement is also suggested to change the upwelling in the northern portion of the nBUS (Jarre et al. 2015).

*Response: We have now discussed the uncertainty in the response of coastal upwelling to the meridional migration of the SAA in lines 67-70 "Enhanced coastal offshore transport was observed in the sBUS when the SAA shifted to the south. However, it is still unclear whether the coastal upwelling across the nBUS is significantly influenced by the meridional displacement of the SAA (Jarre et al., 2055; Lamont et al., 2018)".*

L106: In the Supplementary material the seasonal cycle etc. is analyzed. Is there any support from independent data that ERA5 reliably describe trends in the region? As the paper is dealing with long-term variability, the provided validation is not that meaningful. Please provide an additional comparison of trends and decadal variability between ERA5, ASCAT and SLP. It is well known that such long-term changes are particular uncertain and this study could contribute to assess these uncertainties.

*Response: We computed the long-term trend for both data sets and the 11-year moving average for the St. Helena SLPs (please see Fig. S1 and S6). We have now discussed the results in the main text. Please see lines 166-178 "Consistent with a previous study (Belmonte Rivas and Stoffelen 2019), ERA5 for the southwest African coast agrees well with available observation-based data sets (Supplementary Info; Fig. S1-6). For St. Helena Island, the time series of the monthly, yearly, and 11-year running SLP means obtained from ERA5 evolved very closely with those derived from the observation (Fig. S1). The time series of simulated and observed yearly mean SLPs display only marginal trends, which are about 0.0025 and -0.0056 hPa/year, respectively. The available time series of surface pressure observed at Luanda, Benguela, Port Nolloth, and Cape Town climate stations evolve closely with the EAR5 SLP (Fig. S2). However, the surface pressure time series for all stations is not continuous or sometimes appear to have offset accuracy issues (e.g., Benguela; see Fig. S2b), which do not allow the evaluation of long-term trends. The spatial pattern of the ERA5 and ASCAT meridional wind trends for 2008-2021 agree reasonably well (Fig. S6). Overall, meridional wind near the coastal area of the BUS underwent an upward trend over the last decade. The spatial structure of the trend in the model appears to be smooth relative to that derived from ASCAT wind. In addition, the wind intensification south of 30°S is pronounced in the ASCAT winds, whereas a relatively small trend is derived from ERA5 data".*

L118: volumes transport per unit length ($m^2/s$)

*Response: Corrected. Please see line 187.*

L140: remove comma after coast

*Response: Corrected.*

L149-150: unclear: upwelled (or downwelled) … transported offshore (or onshore) ?

*Response: We have rephrased the sentence to (line 230-231) "The accumulated amount of upwelled water, $W_{total}$, driven by the meridional wind between the coast and the position x is finally transported offshore with the zonal Ekman transport".*

L151: maybe include: i.e., where the meridional wind stress is maximum

*Response: We have added that. Please see lines 231-233 "In this study, the integrals were carried out from the coast up to a point far offshore where the long-term average of wind stress curl equals zero, i.e., where the meridional wind stress is maximum (Fig. 1b)".*

L154: I do not understand what the authors want to say. What is the reason for the simplifications? How good are the made assumptions? How large is the error, e.g., neglecting the onshore geostrophic transport? It would be good to give some more insight.

*Response: It is a valid comment. In our previous study (https://doi.org/10.1175/JPO-D-20-0297.1), using a high resolution ocean model (~5 km over the BUS), we found that the intense upwelling near the coast associated with the divergence of offshore transport is the dominant factor across the entire BUS. In addition, despite the vigorous mesoscale and sub-mesoscale activities in the open-ocean domains, the total amount of upwelled water is very consistent between the model output and the analytical theory. We have further clarified this point in lines 215-225 "Ocean dynamics is associated with many other flow elements, such as the formation of horizontal pressure gradients from upwelling, coastal jets, thermal fronts, sub-mesoscale instabilities, etc. (Fennel 1999; de Szoeke and Richman 1984; Abrahams et al., 2021). For example, the presence of geostrophic onshore directed current can remarkably alter the structure of coastal upwelling (Marchesiello and Estrade, 2010). When the surface and deep Ekman layers overlap over shallow*

*continental shelves, the cross-shore width of coastal upwelling is proportional to the inverse of the bottom slope (Marchesiello and Estrade, 2010). The occurrence of wind drop-off over this zone can cause a large divergence in offshore transport which can significantly alter the structure of coastal upwelling. Despite the importance of the processes above on the local scale, using a high-resolution ocean model, Bordbar et al. (2021) showed that the dominant driver of upwelling near the coast is the divergence of offshore transport. In addition, they showed that the WSCD upwelling could be a reliable estimation of the total amount of upwelled water offshore. This motivates the choice of the atmosphere variables".*

*In addition, we introduced the concept of potential upwelling which indicates upwelling processes driven solely by the vertical flux of atmospheric horizontal momentum into the ocean. We described that in lines 109-118 "We introduce the concept of potential upwelling to distinguish the quantities used in our analysis from those describing realistic, highly complex vertical transport processes in the ocean. In this sense, potential curl-driven upwelling is the upwelling that would take place within an unbounded ocean with only wind-driven surface currents under the absence of other drivers of upwelling, baroclinicity, bottom topography, coastlines, geostrophic flow, inertial or planetary waves. Potential coastal upwelling characterizes an upwelling process driven solely by the alongshore wind and is related only to the cross-shore divergence of the wind-driven cross-shore directed flow. We derive the potential upwelling quantities from analytical theories of ocean dynamics, the steady state Ekman theory, and a theory of coastal upwelling given by Fennel (1999). This keeps the focus on well-defined quantities, even if they do not reflect realistic upwelling that may be modified by other processes like alongshore wind variability, coastally trapped waves, frontal dynamics, etc.".*

Table 1: "alongshore-driven" should be "alongshore-wind-driven"

***Response:*** *Corrected.*

L216: Last sentence is not necessary and distract.

***Response:*** *We have now deleted this sentence.*

L231: It would be good to provide separate correlations for correlations on interannual and decadal timescales. Which timescale dominate the correlation?

*__Response:__ This is a constructive comment from the reviewer. We have now computed the correlation for the yearly mean and 11-year running mean time series. Results are displayed in Figures 1-2 (below). The general pattern of the correlation is similar to that obtained from the monthly mean anomalies (with climatological monthly mean subtracted). Expectedly, using yearly and 11-year running mean time series yields a larger correlation coefficient. However, given the shorter time series in the yearly mean and 11-year running mean time series, the statistical confidence is much lower than the monthly time series. Hence, we think there is merit in not interpreting the yearly mean, and 11-year running mean in this manuscript. In our future research, we want to disentangle the contribution of short- and long-term fluctuation by using extended ERA5, which covers 1940 to the present (~ 8 decades).*

[Figure]

Figure 1: Same as Figure 2 in the main text, but the correlation is computed from the yearly mean time series.

[Figure]

Figure 1: Same as Figure 2, but the correlation is computed from the 11-year running mean time series.

Fig. 2a-c: Please show the complete functions for positive and negative correlations and provide the significance.

*Response: We have now implemented this comment. Please see figure 2 (also see Fig. S9-10) in the new version of our manuscript.*

L280: (used boxes are marked in Fig. 2e)

*Response: Done. Please see line 346 "The differences between the SLP over the SAA core and the areal-averaged SLPs over the nBUS and sBUS (used boxes are marked in Fig. 2e) …".*

L292: "For example, the years 1997 and 2006 are characterized by persistent eastward SAA displacements": I cannot see this behavior in Fig. 3b.

*Response: Sorry for the confusion. It was mainly due to the orientation of labels on the x-axis, which was a bit misleading. We replotted Figure 3 with x-tick-labels normal to the axis. We think the mentioned behavior is clear now.*

L306: "changes in the global heat budget": Do you mean global warming or net radiative imbalance at the top of the atmosphere? Why are 1990s important the increase started earlier.

*Response: This is correct. We intend to discuss the decadal rate of global warming over the 1990s, which was large than in previous/next decades. If the mechanism suggested by Bakun is a dominant factor, it should, to some extent, appear in the rate of global warming over the 1990s. However, the SAA time series doesn't show that. We have now rephrased the sentence to (lines 375-376) "Despite the accelerated rate of global average temperature over 1990s (Bordbar et al., 2019), the SAA intensity and position remained steady and underwent no significant trend."*

L309: "due to the enhanced radiative forcing": what is meant with radiative forcing? of the ocean? Do you mean global warming?

*Response: We have now changed "due to the enhanced radiative forcing" to "due to global warming". Please see lines 378-379 "If there is any tendency in the intensity and location of the SAA due to global warming, it is presumably too small to emerge from background climate fluctuations".*

L313: "positive trend is more prominent over higher latitudes": Is this associated with poleward migration of the center of the SAA?

*Response: Since the spatial pattern of the trend displays rather a meridional structure and shares many similarities with the Positive phase of the Southern Annular Mode (SAM; https://doi.org/10.1002/wcc.652). We conjecture this changes can be largely attributed to long-term variations of the SAM. Please see lines 381-386 "In general, the size of the trend varies more in the meridional direction and is more prominent over higher latitudes, particularly for July-October. The most prominent trend is found in the southwest and the southeast of the domain in Jul-Oct (Fig. 4c) and Jan-Apr (Fig. 4b), respectively. The structure of the trend reminds the recent multi-decadal trend in the SAM, which is associated with an enhanced meridional SLP gradient between the polar and mid-latitudes (Wachter et al., 2020; Fogt & Marshall, 2020)".*

L365: "Based on equation 5, …" In equation 5, upwelling velocity depend on x. Do you mean maximum upwelling velocity calculated using R1 given in Fig. S5? … corresponds to a maximum upwelling velocity …

*Response: In fact, we estimated the upwelling velocity when x=0 in the equation 5. We have now mentioned that in line 336 "Based on equation 5, this corresponds to a maximum upwelling velocity (i.e., x=0) of about 6.9 m/d".*

L397: the spatial resolution

*Response: Done. Please see "the spatial resolution" line 474.*

L408-413: this section must be improved: possible there are multidecadal changes or trends and there are interannual to decadal variability. It is not obvious from the analysed data what is due to global warming or what is internal climate variability. Used terms such as "historical changes" or "historical trends" are unclear. These conclusions must be formulated more carefully.

*Response: We agree with the reviewer. We have now re-written this paragraph. Please see lines 486-493 "Despite a slight upward SLP trend in the subtropical South Atlantic during 1979-2021, the ratio between changes associated with the long-term SLP trend (i.e., Δ) and the standard deviation of the long-term trend subtracted yearly mean SLP is small across the entire domain. Further, potential upwelling quantities, including $W_{coast}$ and $W_{curl}$, in several upwelling cells remained steady and exhibited neither a significant long-term trend nor prominent changes in the characteristics of the variability (i.e., period, amplitude, and extremes). Overall, our results neither demonstrate nor rule out the potential impacts of anthropogenic global warming on the atmospheric drivers of upwelling in the BUS. A possible explanation is that a much longer time is likely required to detect the robust global warming signals in the wind-driven upwelling across the BUS".*

---

## Author Comment (AC2)

We thank Dr. Fabien Desbiolles for his constructive comments. We overall agree with the points raised, which have been considered in revising the manuscript. In the following, our responses to the reviewers are shown in *blue italics*.

The reviewer's comments encouraged us to introduce the concept of potential upwelling to clarify our methodology. Please see lines 108-118 "We mainly analyze the variability of atmospheric quantities and associated impacts on upwelling across the BUS. We introduce the concept of potential upwelling to distinguish the quantities used in our analysis from those describing realistic, highly complex vertical transport processes in the ocean. In this sense, potential curl-driven upwelling is the upwelling that would take place within an unbounded ocean with only wind-driven surface currents under the absence of other drivers of upwelling, baroclinicity, bottom topography, coastlines, geostrophic flow, inertial or planetary waves. Potential coastal upwelling characterizes an upwelling process driven solely by the alongshore wind and is related only to the cross-shore divergence of the wind-driven cross-shore directed flow. We derive the potential upwelling quantities from analytical theories of ocean dynamics, the steady state Ekman theory, and a theory of coastal upwelling given by Fennel (1999). This keeps the focus on well-defined quantities, even if they do not reflect realistic upwelling that may be modified by other processes like alongshore wind variability, coastally trapped waves, frontal dynamics, etc." and lines 465-468 "We introduced the concept of potential upwelling which indicates upwelling processes driven solely by the vertical flux of atmospheric horizontal momentum into the ocean. This approach helps to focus on well-defined quantities, even though they do not reflect a comprehensive picture of processes that can considerably modify the upwelling (i.e., alongshore wind variability, coastally trapped waves, frontal dynamics, etc.)".

**Reviewer #2 (Dr. Fabien Desbiolles)**

The paper discusses the relationship between long-term trends in the position and strength of the South Atlantic Anticyclone (SAA) and upwelling dynamics. The author uses ERA5 reanalyses and focuses on different aspects of the upwelling system, namely the windstress-driven and the curl-driven upwelling. Overall, the paper is well presented and well written. In particular, the methodology is clear and seems promising to disentangle different aspects of the complex and multiscale nature of an Eastern Bounduary Upwelling System (EBUS), i.e. from large-scale wind forcing to local-scale modifications of the wind by orography and/or SST/wind interactions.

In particular, the authors show that a robust change in the position and strength of the SAA together affects both curl-driven and wind-driven upwelling, with a pronounced increase in the curl-driven mechanism. However, this robust change cannot be attributed to Bakun's theory because it is difficult to attribute the SAA change to climate mode variability or to the difference in warming between the land and ocean. Strong spatial heterogeneity of a weak long-term trend in ERA5 is shown, making any strong conclusion difficult. However, the limitations of the study and the use of ERA5 are well presented in the last sections of the paper. The paper could be suitable for publication after discussion of the following aspects:

(1) ERA5 is a medium-resolution reanalysis with a relatively short time span. It remains difficult to relate anthropogenic global warming to the long-term trends present in the data set. Also, with an analysis period of 40 years, it is complicated to disentangle the anthropogenic signal from climate mode and decadal variability. I remain convinced that a lot can be done and said with ERA5, which remains a very useful dataset, albeit limited. For example, one can ask whether there is a signal in ERA5 for differential warming over land and ocean. If so, can this signal explain the long-term trend in SSA strength and position? How are climate modes represented in the ERA5 dataset, and what are the consequences for SAA position and strength? The paper gives some hints, but would benefit from a clearer and more thorough discussion of these two aspects. Please note that this comment does not necessarily require important new diagnostics.

*Response: We computed the linear trend in the ERA5 surface air temperature over 1979-2021, and the result has been displayed in Figure S11. As can be viewed in this figure, the rate of warming*

*over land is more significant than that in the adjacent ocean. We discussed that in the main text, lines 348-351 "In general, the rate of warming over land is larger than the adjacent ocean across the BUS in 1979-2021 (Fig. S11). This feature is more pronounced in the BUS northernmost sector (i.e., Cape Frio and Kunene upwelling cells), where the rate of warming over land exceeds 0.05 °C/year. Thus, one may expect an enhanced SLP gradient between the land and the adjacent ocean".*

*In addition, we have now addressed the ability of ERA5 reanalysis to represent the impacts of dominants of climate modes (Atlantic Niño, ENSO, SAM) on the SAA. Please see lines 134-140 "The connection of the SAA variability with the onset and development of Atlantic Niño events and associated SST changes over the Angola Benguela front (15°S-17°S) is well represented in the ERA5 data (Prigent et al., 2020). The poleward (equatorward) displacement of the SAA during austral summer when the tropical Pacific features La Niña (El Niño) events is well represented in the ERA5 data sets (Rouault & Tomety, 2022). In addition, the spatial structure of the SAM and meridional wind anomalies over the southern hemisphere during different SAM phases are well reproduced in the ERA5 data sets (Marshall et al., 2022)".*

*We also discuss this point in lines 486-493 "Despite a slight upward SLP trend in the subtropical South Atlantic during 1979-2021, the ratio between changes associated with the long-term SLP trend (i.e., Δ) and the standard deviation of the long-term trend subtracted yearly mean SLP is small across the entire domain. Further, potential upwelling quantities, including Wcoast and Wcurl, in several upwelling cells remained steady and exhibited neither a significant long-term trend nor prominent changes in the characteristics of the variability (i.e., period, amplitude, and extremes). Overall, our results neither demonstrate nor rule out the potential impacts of anthropogenic global warming on the atmospheric drivers of upwelling in the BUS. A possible explanation is that a much longer time is likely required to detect the robust global warming signals in the wind-driven upwelling across the BUS".*

(2) Several mechanisms are responsible for curl-driven cross-shore upwelling.

(i) Offshore, winds blowing along a permanent or semi-permanent front adjust through the so-called downward momentum mixing mechanism, with a decrease (increase) of the surface wind

over the cold (warm) flank of the front (see e.g. Chelton et al., 2004). Even with the relatively low resolution of the dataset, this mechanism is important in the dynamics of the ERA5 surface winds, especially in the EBUS where near-neutral conditions prevail. (ii) Near the coast, the coastal wind dynamics are mainly driven by the differential stress between the ocean and the continent and the role of the coastal orography. This mechanism usually implies a dropoff zone (as mentioned in the paper) and possibly a cape effect. Both points mentioned here are driven by important structures in the curl that favor upwelling and are hidden in the Wcurl calculated in the paper. The paper would benefit from a fuller discussion of these aspects. The orography-induced wind drop is undoubtedly not well characterized in ERA5 due to poor resolution (coastal points also suffer from the so-called Gibbs phenomena). The coastal orography varies from the southern seasonal upwelling to the more permanent northern counterpart; inducing a different permanent wind curl signal at the coast.

*__Response:__ Thanks for this constructive comment and for introducing this study. We read the suggested paper and cited it in the new version of our manuscript. We have now discussed the uncertainty in the ERA5 data set and briefly discussed the suggested mechanism by the reviewer. Please find our response to this comment in lines 144-154 "There is another source of data uncertainty over coastal areas with frequent upwelling events. The offshore transport of upwelled cold water often forms SST fronts near the coast (de Szoeke and Richman 1984), altering the local structure of the wind stress curl. On the cold side of the front, the near-surface air column is stabilized, decelerating the local wind speed. At the same time, the air column is destabilized over the warm side, and the wind intensifies (Chelton et al., 2004). In this way, small-scale SST fronts drive local convergence and divergence of the surface wind, which is proportional to the size of the crosswind SST gradient. This aspect of small-scale (i.e., sub-mesoscale) ocean-atmosphere interaction is not adequately represented in ERA5 reanalysis because the model's resolution is too coarse. Further, orographic features near the coast (i.e., mountain passes, coastline geometry, and capes), which are not well resolved in the model used in ERA5 reanalysis, can diverge or converge the winds locally and alter the structure of wind stress curl (Chelton et al., 2004)".*

*We also mentioned these point in line 472-475 "Since it provides an accurate estimate of $W_{total}$, this approach is promising and suitable for the ERA5 data, even though the spatial resolution of*

*this data is too coarse (i.e., 0.25°×0.25°) to resolve the small-scale processes driven by coastal SST fronts, orographic features near the coast, which can alter the structure of surface winds and amplify or reduce the coastal wind drop-off.".*

*And lines 496-499 "Indeed, one cannot attribute the entire wind field variability over the BUS solely to the SAA and the regional cross-shore surface air temperature gradient. Localized drivers of the surface winds (e.g., sub-mesoscale SST fronts, orographic effects, eddies, and land-sea breezes), which the operational model used in ERA5 reanalysis does not resolve, may significantly alter the surface wind field".*

The wind/SST interaction is an important factor for the Ekman pumping in the BUS. Even though it is a coupled mechanism, some insight into the SST forcing used in ERA5 would be nice to discuss (possible cold biases; structures of the fronts, etc.). The paper, and with legitimate argument, examines only atmospheric variables, but the quality of the SST forcing seems crucial to discuss, especially with the role of the front in shaping the Ekman pumping.

***Response:*** *We agree with the point raised by the reviewer. We briefly discuss the data used for the data assimilation in ERA5 and how the SST-wind interaction is represented in this reanalysis. We also briefly discuss the source of data uncertainty. Please see lines 126-133 "HadISST2 data set, which was developed by the UK Met Office Hadley Centre, is widely implemented in ERA5 reanalysis (Hirahara et al., 2016). HadISST2 is on a 0.25°×0.25° regular grid and is derived from in-situ observations and two infrared radiometers, including the Along Track Scanning Radiometer (ATSR) and the Advanced Very High-Resolution Radiometer (AVHRR) (Hirahara et al., 2016). From mid-2008 onward, OSTIA SST from the UK Met Office with a resolution of 0.05°×0.05° was also used in ERA5 reanalysis (Hirahara et al., 2016). OSTIA is based on various types of observation, including in-situ observation, geostationary satellites and microwave imagers. In the Agulhas region located on the southern border of the BUS, both OSTIA and HadISST successfully represent the sub-mesoscale eddies."*
*We also discuss the potential impacts of SST fronts on the wind stress and the wind stress curl. please see lines 144-154 (please see previous comment).*

(3) The morphology of the continental shelf is important when comparing WSD and WSCD upwelling. As Marchesiello and Estrade (2010), the competition between reducing coastal upwelling and increasing Ekman pumping can be assessed by comparing the length scales of coastal upwelling (Lu) and orographically induced wind drop-off (Ld). If the two length scales are equal, there is no effect on the total upwelling water in the cross-shore direction. The continental shelf of the area, especially around the upwelling cells mentioned, could be discussed and drawn on one of the maps.

*Response:* Thanks for informing us of the study by Marchesiello and Estrade (2010) entitled "Upwelling limitation by onshore geostrophic flow". The author of this paper used an analytical theory alongside the regional ocean model, which provide very interesting results. As the reviewer correctly pointed out, the size of Lu is in the order of several kilometers, which ERA5 data cannot capture. In addition, information on wind drop-off properties suffers from the same issue. We wanted to implement this comment to more precisely evaluate the length scale of coastal upwelling and wind drop-off. However, the spatial resolution of ERA5 data introduces a barrier to correctly estimating the factors mentioned above. We also found the discussion of the suggested study around the superposition of onshore geostrophic current and along-shore driven upwelling very interesting. We discuss the result of this study in line 215-221 *"Ocean dynamics is associated with many other flow elements, such as the formation of horizontal pressure gradients from upwelling, coastal jets, thermal fronts, sub-mesoscale instabilities, etc. (Fennel 1999; de Szoeke and Richman 1984; Abrahams et al., 2021). For example, the presence of geostrophic onshore directed current can remarkably alter the structure of coastal upwelling (Marchesiello and Estrade, 2010). When the surface and deep Ekman layers overlap over shallow continental shelves, the cross-shore width of coastal upwelling is proportional to the inverse of the bottom slope (Marchesiello and Estrade, 2010). The occurrence of wind drop-off over this zone can cause a large divergence in offshore transport which can significantly alter the structure of coastal upwelling".*

---

## Author Response (AR2)

We thank the reviewer for raising very constructive comments. We overall agree with the points raised, which have been implemented in revising the manuscript. In the following, our responses to the reviewer are shown in italics.

**Response to Reviewer #1**

L6: "upwelling near the southwest African coasts is primarily alongshore-wind-driven". This is strictly not correct as the Angolan upwelling that is also located along the southwest African coast is driven by coastally trapped waves in combination with tidal mixing on the shelf.

> *Response: This is a valid point. We have now rephrased this sentence to (lines 6-8) "Like other Eastern Boundary Upwelling Systems, in the Benguela Upwelling System, the upwelling along the coastline is primarily alongshore-wind-driven. In contrast, it is mainly driven by the wind stress curl farther offshore".*

L7: "primarily alongshore-wind-driven, whereas it is controlled mainly by the wind stress curl farther offshore" It is unclear what is meant: later you use alongshore-wind-driven coastal upwelling. Could be written more clearly. What is the difference between "driven" and "controlled". Does it correspond to mean upwelling and its variability? Please clarify.

> *Response: it is correct. We have now changed the sentence to (lines 6-8) "Like other Eastern Boundary Upwelling Systems, in the Benguela Upwelling System, the upwelling along the coastline is primarily alongshore-wind-driven. In contrast, it is mainly driven by the wind stress curl farther offshore".*

L12: "closed to the coastal band" I would suggest: "within the coastal zone".

> *Response: corrected. Please see line 12.*

L13: I would suggest "for both coastal upwelling and wind-stress-curl-driven upwelling" or "for both alongshore-wind-driven coastal upwelling and wind-stress-curl-driven upwelling".

> *Response: Done. Please see lines 13-15 "Even though the detailed structure of surface wind over the coastal zone matters for both alongshore-wind-driven coastal upwelling and wind-stress-curl-driven upwelling, we show that it is not of major importance for the total amount of upwelled water".*

L15: anticyclone

> *Response: Corrected. Please see line 15.*

L15: zonally integrated

*Response:* we have now changed "integrated" to "zonally integrated". Please see line 15.

L16: such connection

*Response:* we have now changed "this connection" to "such connection". Please see line 16.

L21-23: Do you want to say that there are no multi-decadal trends? Supported by a non-existing multi-decadal trend? Maybe: This view is further supported by the coastal and wind-stress-curl-driven upwelling in several upwelling cells showing hardly any significant multi-decadal trends.

*Response:* we rephrased the sentence accordingly. Please see lines 22-23 "This view is further supported by the coastal and wind-stress-curl-driven upwelling in several upwelling cells showing hardly any significant multi-decadal trends".

L53: excites downwelling Kelvin waves

*Response:* Done. Please see line 53 "…which excites downwelling Kelvin waves propagating …".

L187: I would suggest to remove "vertically integrated" as the transport is not integrated, but the velocity. Better use Ekman zonal and meridional volume transport per unit length.

*Response:* we have now rephrased the sentence to (line 187) "In this theory, Ekman zonal ($U_E$) and meridional ($V_E$) volume transport per unit length ($m^2/s$) are expressed as".

L210 tau^y is already explained above

*Response:* corrected. We changed the sentence to (line 210) "Here, x is the distance to the coast".

L295: alongshore

*Response:* we changed (line 295) "along-shore" to "alongshore".

L374: yearly and the monthly means

*Response:* corrected. Please see line 374 "the yearly and the monthly means".

L374: Dashed lines

*Response:* corrected. Please see line 374 "Dashed lines represent …"

L397: obtained for yearly mean SLP

*Response:* corrected. Please see line 397.

L412: obtained for yearly mean SLP

*Response:* corrected. Please see line 412.

Figure S3: define R

*Response:* sorry for the confusion we caused. We have now rephrased the last sentence of figure S3 caption to "The linear correlation (R) between the ASCAT and ERA5 meridional winds is represented in the bottom-right corner of panels b-h".

Figure S4: please provide units

*Response:* we have now added the unit of meridional component of wind speed, which is m/s.

Caption Figure S5: ERA-5

*Response:* corrected.

Figure S7: There are no red dots. Please correct.

*Response:* we have now corrected the figure caption. It is now "Circles indicate the $R_1$ from Chelton et al. (1998), and dashed line represents the interpolated $R_1$ that is used in this study".